evolution, ecology

sexually transmitted diseases, microbiome, reproductive ecology

**Author for correspondence:**
Sara Bellinvia
e-mail: sara.bellinvia@uni-bayreuth.de

# Mating changes the genital microbiome in both sexes of the common bedbug *Cimex lectularius* across populations

Sara Bellinvia[1], Paul R. Johnston[2], Susan Mbedi[3,4] and Oliver Otti[1]

[1]Animal Population Ecology, Animal Ecology I, University of Bayreuth, Universitätsstraße 30, 95440 Bayreuth, Germany
[2]Institute for Biology, Free University Berlin, Königin-Luise-Straße 1-3, 14195 Berlin, Germany
[3]Museum für Naturkunde, Leibniz Institute for Evolution and Biodiversity Research, Invalidenstraße 43, 10115 Berlin, Germany
[4]Berlin Center for Genomics in Biodiversity Research (BeGenDiv), Königin-Luise-Straße 1-3, 14195 Berlin, Germany

 SB, 0000-0002-6311-3112; PRJ, 0000-0002-8651-4488

Many bacteria live on host surfaces, in cells and in specific organ systems. In comparison with gut microbiomes, the bacterial communities of reproductive organs (genital microbiomes) have received little attention. During mating, male and female genitalia interact and copulatory wounds occur, providing an entrance for sexually transmitted microbes. Besides being potentially harmful to the host, invading microbes might interact with resident genital microbes and affect immunity. Apart from the investigation of sexually transmitted symbionts, few studies have addressed how mating changes genital microbiomes. We dissected reproductive organs from virgin and mated common bedbugs, *Cimex lectularius* L., and sequenced their microbiomes to investigate composition and mating-induced changes. We show that mating changes the genital microbiomes, suggesting bacteria are sexually transmitted. Also, genital microbiomes varied between populations and the sexes. This provides evidence for local and sex-specific adaptation of bacteria and hosts, suggesting bacteria might play an important role in shaping the evolution of reproductive traits. Coadaptation of genital microbiomes and reproductive traits might further lead to reproductive isolation between populations, giving reproductive ecology an important role in speciation. Future studies should investigate the transmission dynamics between the sexes and populations to uncover potential reproductive barriers.

## 1. Introduction

Animals have intimate associations with bacteria. Even reproductive organs and the semen of a variety of animals often harbour several different microbe species [1–9]. In insects, many studies have investigated intracellular microbes that manipulate host reproduction, among others by male killing, or cytoplasmic incompatibility (reviewed in [10]). Besides these reproductive manipulators, extracellular microbes have been found on the male copulatory organ and within the female reproductive organs (reviewed in [11]; also see [12,13]). Interestingly, the microbiomes of whole body homogenates or the gut [14–16] and even the genital microbiomes [3,12,13] are sex-specific in a variety of species. The exact reasons for those differences as well as the role of genital microbes in reproduction and the exact composition of these genital microbiomes are unknown, especially for insects.

Microbiomes are dynamic and react to environmental change, such as diet [17,18], climate [19] and time of day [20]. In humans, life-history events such as pregnancy [21] and menopause [22] change the vaginal microbiome. Another life-history event with the potential to affect genital microbiomes is mating as

the microbiomes of both sexes encounter each other and potentially interact. Mating-induced alterations of the genital microbiome are documented in vertebrates, for instance in birds [23] and humans [24,25]. By contrast, little is known about the potential effects of mating on the composition of genital microbiomes in invertebrates.

Microbes are sexually transmitted in a large range of species (e.g. [26–28]). Not only microbes that cause sexually transmitted infections (STI) are transferred during mating. We assume also environmental bacteria in and on the genitalia [13,29,30] can be transferred or enter the reproductive organs via genital openings or copulatory wounds, which frequently occur in invertebrates [31]. The environmental bacteria with the potential to colonize the genitalia because they can tolerate the local abiotic factors, such as pH or temperature, will henceforth be called opportunistic microbes (OM). OM do not necessarily always cause an infection but become pathogenic when the host immune system is disturbed [32].

Once transferred into the female genital tract, sperm is exposed to a rich microbial flora [1,5]. In this context, OM might have a more direct effect compared to microbes causing STI: OM can decrease sperm motility and agglutinate spermatozoa in humans [33–37] and increase sperm mortality *in vitro* in insects [38]. Therefore, we predict male reproductive success to depend on the microbe communities within the female. In addition, invading OM might disturb the present genital microbiome. Because sexual intercourse decreases the relative abundance of one of the dominant species in the vaginal microbiome [24], similar mating-induced disturbances of insect genital microbiomes are conceivable. Copulatory wounding during mating further increases the risk of OM invasion in invertebrate species [39,40] and in humans [41]. Sexually transmitted bacteria disturbing the composition of the genital microbiome have been predicted to select not only for a host immune response to prevent uncontrolled growth [11], but also for defensive responses in the resident microbiota [12]. Therefore, the host and its endosymbionts have a mutual interest in keeping invading bacteria in check. In some organisms, the resident microbiota is part of the interaction with invading microbes [42–45].

If genital microbiomes are subject to OM, an adaptation to environmental microbes seems conceivable. North American women from four different ethnicities harboured distinct vaginal microbiomes [7], raising the question whether there are conserved differences in the community composition between populations and whether mating-induced changes are involved in this differentiation.

Bedbugs are an interesting system to study mating-induced changes of insect genital microbiomes because several organs are involved in reproduction. During mating and the ejaculate transfer, microbes potentially invade these reproductive organs. The male ejaculate consists of spermatozoa that are stored in the male sperm vesicles and seminal fluid from the seminal fluid vesicles. After mounting the female, the male transfers the ejaculate via its copulatory organ, the paramere, into the female paragenital copulatory organ, the mesospermalege [46]. After a few hours, sperm travel through the haemolymph towards the ovaries [46]. All tissues involved could be invaded by microbes that impose a risk of infection or sperm damage.

Microbial communities differ between reproductive organs in bedbugs [13]. Copulation increases the similarity of female and male organs, and bacteria present in mated but not virgin individuals of one sex are found in the opposite sex. Also, some of the resident bacteria are replaced with introduced bacteria [13]. Our aim is to investigate whether these findings within one bedbug population are a general pattern across bedbug populations and whether genital microbiomes differ between populations. Here, we use a community ecology approach based on 16S rRNA sequencing data from the genital microbiomes (i.e. the microbial communities of all external and internal organs involved in reproduction) of four bedbug populations. We focus on a potential difference in the microbial community between populations, between the two sexes, between organs, and between virgin and mated individuals. We thereby evaluate the effect of differences in environmental microbes and genotypes between populations on the genital microbiome and the risk of a change in composition via copulation.

## 2. Material and methods

### (a) Bedbug culture

We used four large stock populations of the common bedbug (*Cimex lectularius* L.) out of which three populations were field caught from London, UK in 2006 (A), from Nairobi, Kenya, in 2008 (B), and from Watamu, Kenya, in 2010 (C). The fourth population (D) was a long-term laboratory stock originally obtained from the London School of Hygiene and Tropical Medicine over 20 years ago. All populations were held in separate 60 ml plastic pots containing filter paper in a climate chamber at $26 \pm 1°C$, 70% relative humidity, and a light cycle of 12 L : 12D at the University of Bayreuth and fed weekly with the same sterile food source using the protocol of Hase [47]. After eclosion, all individuals were kept in sex-specific groups of 20–30 individuals in 60 ml plastic pots containing filter paper. Males were fed twice. Females were fed three times, with the last feeding on the day of dissection because fully fed females cannot resist copulation [48].

### (b) Mating and sample preparation

Dissections and DNA extractions were conducted at the University of Bayreuth in 2016 and 2017. We dissected 643 three-week-old males and females from four populations (population A: $n = 163$, population B: $n = 160$, population C: $n = 160$, population D: $n = 160$). Half of the individuals were randomly mated before dissection. For this, females were placed individually in a petri dish with a fresh filter paper. Then a male was added. Sixty seconds after insertion of the paramere, female and male were separated with forceps and transferred to separate containers. Both were dissected 1–2 h after successful mating, ensuring the sperm were still inside the mesospermalege of mated females at the time of dissection. The potential of contamination was minimized by a laboratory butane burner (Labogaz 206, Campingaz, Hattersheim, Germany) placed next to the dissection microscope. The dissection kit was autoclaved each day and all forceps and surgical scissors were dipped in ethanol (70%) and flame-sterilized before each dissection.

We collected different reproductive tissues and cuticle samples from both sexes ($n = 10 \pm 0$ per mating status, organ, sex and population; mean ± s.d.; see electronic supplementary material, table S1). Each sample was taken from a different bedbug since it is difficult to obtain all tissues from the same individual. From males we collected sperm vesicles, seminal fluid vesicles and paramere. In females, we investigated the mesospermalege, ovaries and haemolymph. Haemolymph was collected using a sterilized glass capillary pulled to a fine point. Each tissue was transferred into an Eppendorf tube containing 150 µl of phosphate-buffered saline. We sampled the cuticle by transferring whole females or males, whose paramere had been removed, into a tube. As a

contamination control, we put an open Eppendorf tube containing only phosphate-buffered saline next to the dissection microscope during dissections. This tube was processed in the same way as all tissue samples. All samples were frozen in liquid nitrogen and stored at −80°C.

## (c) Molecular methods

Prior to DNA extraction, we homogenized the samples using pestles made from sterile pipette tips (200 µl). We followed the protocol of the MO BIO UltraClean Microbial DNA Isolation Kit (dianova GmbH, Hamburg, Germany), which includes a bead beating step, except that we dissolved the DNA in 30 µl elution buffer for higher yield. DNA was stored at −20°C. To control for contamination during DNA extraction, we performed one extraction without adding any tissue.

The library preparation and sequencing were done in the laboratory of the Berlin Center for Genomics in Biodiversity Research. The samples were split up into four sequencing runs, each balanced in terms of population, sex, organ and mating status, resulting in four libraries with $128 \pm 25$ (mean $\pm$ s.d.) samples each, including controls (see below). Using the universal primers 515fB (5′-GTGYCAGCMGCCGCGGTAA-3′ [49]) and 806rB (5′-GGACTACNVGGGTWTCTAAT-3′ [50]), we amplified the variable V4 region from the bacterial 16S rRNA (denaturation: 94°C, 3 min; 30 cycles of denaturation: 94°C, 45 s, annealing: 50°C, 1 min, extension: 72°C, 90 s; extension: 72°C, 10 min). After amplicons had been purified using Agencourt AMPure XP beads (Beckmann Coulter GmbH, Krefeld, Germany), a unique combination of two eight nucleotide long index sequences were used for barcoding each sample in a second PCR (denaturation: 95°C, 2 min; 8 cycles of denaturation: 95°C, 20 s, annealing: 52°C, 30 s, extension: 72°C, 30 s; extension: 72°C, 3 min). After another purification step with AMPure beads, the DNA concentration of the PCR products was quantified with the Quant-iT PicoGreen dsDNA Assay Kit (Invitrogen, Carlsbad, CA, USA). Samples that had a higher concentration than $5 \, \text{ng} \, \mu\text{l}^{-1}$ were diluted to this concentration; all other samples were left undiluted. The quality of the pooled amplicons was verified with a microgel electrophoresis system (Agilent 2100 Bioanalyzer, Agilent Technologies, Santa Clara, CA, USA). The resulting libraries were subjected to an Illumina MiSeq sequencer and paired-end reads were generated using PhiX. For each plate and PCR type, we had 1–2 negative controls containing only purified water instead of DNA, resulting in 16 controls for the target PCR and 13 controls for the indexing PCR across all sequencing runs. Four additional samples per sequencing run contained the bacterial DNA from a whole homogenized bedbug to increase sequencing depth.

## (d) Bioinformatical analysis

### (i) Data processing and check for contamination

All data processing and analyses were performed in R (version 5.3.1 [51]) with the packages *dada2* (version 1.10.1 [52]), *decontam* (version 1.2.0 [53]), *DECIPHER* (version 2.10.2 [54]), *phangorn* (version 2.4.0 [55,56]), *phyloseq* (version 1.19.1 [57]), *pairwiseAdonis* (version 0.0.1 [58]), *vegan* (version 2.4-5 [59]) and *edgeR* (version 3.22.1 [60,61]). We used the *dada2* [52] pipeline to filter and trim the sequences. The first 10 bp were removed and the sequences were truncated after 260 bp (forward reads) and 200 bp (reverse reads), or at the first instance of a quality score ≤2. Sequences with expected errors greater than 2 were discarded. The remaining sequences were dereplicated and denoised. We constructed a sequence variant (SV) table and removed chimeric sequences. We scored all controls with the *decontam* package [53] based on prevalence and removed all contaminants (electronic supplementary material, table S2). All SVs that occurred in only one sample and that had less than 0.01% of the total number of unfiltered reads were removed as suggested by Caporaso *et al.* [62]. Out of 21 478 209 reads, 13 129 154 remained in the final dataset. The highest proportion of reads was lost during the first quality filtering step (18%) and decontamination (15%). The taxonomy of the remaining taxa was assigned with the Greengenes database [63]. We verified the taxonomical assignment with Blast2Go [64]. If Blast2Go did not find a match, we used NCBI's BLASTn. In both cases, we excluded uncultured or environmental samples from the database used for taxonomic assignment. The taxonomic assignments of the different algorithms were in accordance for kingdom until genus level in 89 out of 126 SVs. In one of the mismatches, the BLAST hit with the highest e-value and coverage belonged to an endosymbiont of *C. lectularius*, the unclassified gammaproteobacterium mentioned by Hosokawa *et al.* [65]. In five other mismatches, all BLAST hits agreed on one genus. We therefore changed the taxonomy assignment of these SVs. For 31 cases without a clear BLAST result, we kept the Greengenes assignment for the levels that were congruent with the BLAST results. We changed the assignment of all other levels to 'Unclassified'. After aligning sequences with the *DECIPHER* package [54], we used the *phangorn* package [55,56] to fit a maximum likelihood tree (GTR + G + I). We added all information regarding sample type, read numbers for samples and SVs, and taxonomic assignments to the electronic supplementary material, table S3–S5.

### (ii) Statistical analysis

All analyses except for the differential abundance test were based on relative abundances. Comparisons of the microbiome composition were based on Bray–Curtis dissimilarities obtained with the *phyloseq* package [57] as well as the alpha diversity estimates (Simpson index $(1 - D)$). Additionally, we used weighted UniFrac distances to compare the genital microbiomes from virgin and mated bedbugs.

### (iii) Microbiomes of virgin bedbugs

We analysed the differences in microbiome composition between internal reproductive organs (sperm and seminal fluid vesicles of males; mesospermalege, haemolymph and ovaries of females), external reproductive organs (male paramere) and cuticle with a PERMANOVA (999 permutations, *vegan* package [59]) followed by a multilevel pairwise comparison using pairwise PERMANO-VAs (*pairwiseAdonis* package [58]) and Benjamini–Hochberg adjusted *p*-values. Between-individual differences across the three groups were analysed with a multivariate test for homogeneity of group dispersions with the *vegan* package [59]. We compared the genital microbiome composition between populations, sexes and organs using a PERMANOVA with the fixed effects population, sex and organ nested within sex. Between-individual variation was compared with three separate tests for homogeneity of group dispersions for populations, sexes and organs.

### (iv) Mating-induced effects on genital microbiomes

Compositional differences of genital microbiomes from virgin and mated bedbugs were analysed with a PERMANOVA with the fixed effects population, organ, mating status and their interactions. Removing the non-significant interactions improved the AIC. We then compared between-individual variation with separate tests for homogeneity of group dispersions for population, organ and mating status.

We estimated the proportions of bacterial strains present in mated individuals potentially originating from virgin individuals of the opposite sex using the programme SourceTracker2 [66]. To generate the input files, we split the dataset by population and transformed the *phyloseq* data into biom format with the *biomformat* package [67].

With the quasi-likelihood tests in the *edgeR* package [60,61], we tested for differential abundance of SVs, here represented by read

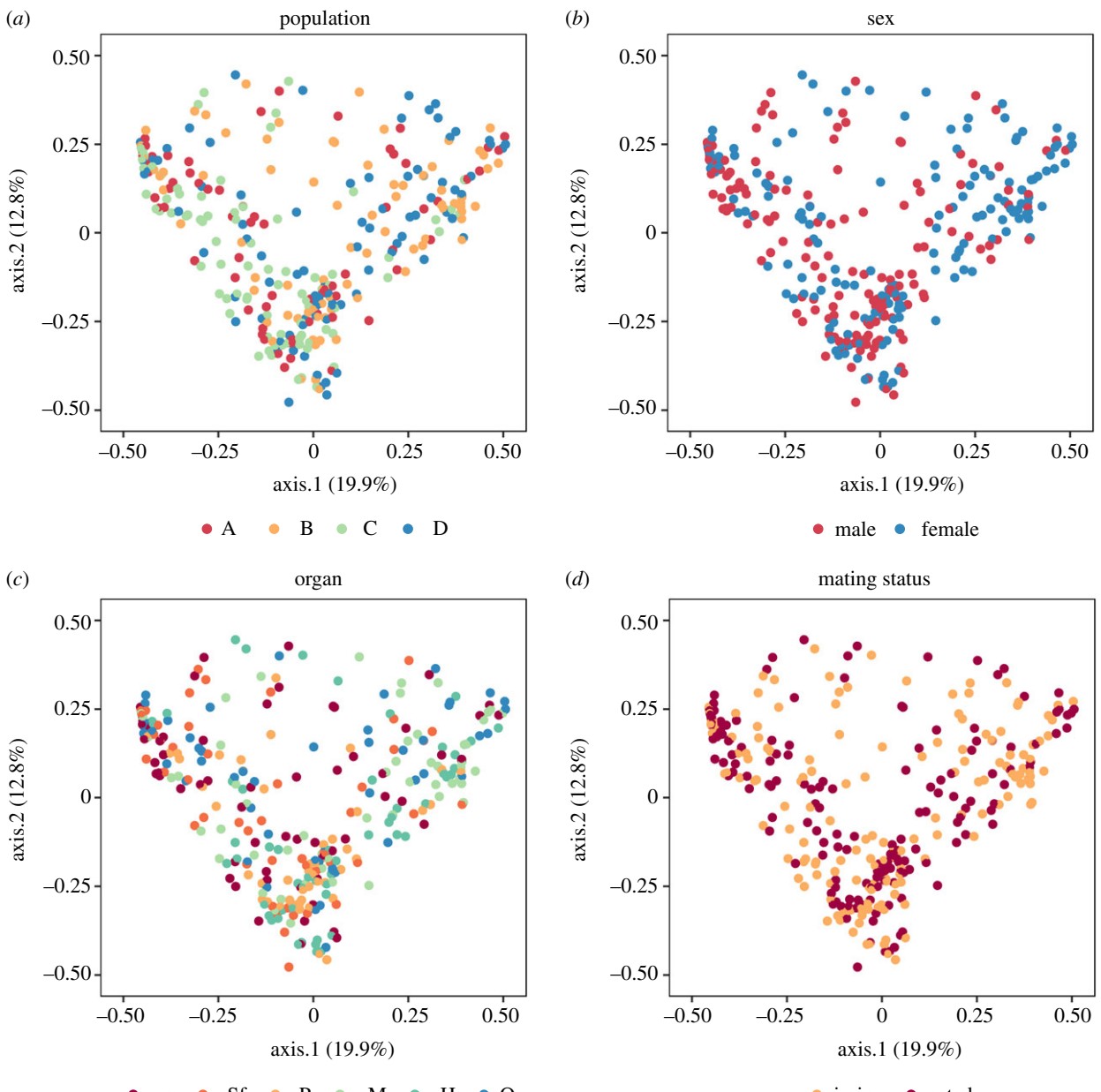

**Figure 1.** PCoA of all groups of genital microbiomes performed with the *phyloseq* package [57] on Bray–Curtis dissimilarities. Organs are sperm vesicle (S), seminal fluid vesicle (Sf), paramere (P), mesospermalege (M), haemolymph (H) and ovary (O). (Online version in colour.)

numbers, between the genital microbiomes of virgin and mated bedbugs. Normalization factors were calculated with the relative log expression [68] and applied to read counts with an added pseudocount of 1. Contrasts were built for every mating status within organ and population. *p*-values were adjusted with the Benjamini–Hochberg procedure (FDR = 0.05). The proportion of SVs with significant differential abundance was analysed with a logistic regression with quasibinomial error structure and the fixed effects population and sex, including their interaction. Groups were then compared with an ANOVA. Non-significant interactions were removed to analyse the main effects, which improved the AIC.

## 3. Results and discussion

We sequenced 643 samples from bedbug reproductive organs or cuticle via 16S rRNA amplicon sequencing to characterize the composition of the genital microbiomes and investigate the effect of mating. We sampled the cuticle from both sexes, the external intromittent organ (paramere) and the internal sperm vesicles and seminal fluid vesicles from males and the

sperm-receiving organ (mesospermalege), the ovaries and the haemolymph, which are all internal female organs. After filtering, we obtained 126 sequence variants (SVs) from 495 samples ($n = 8 \pm 1$ per mating status, organ, sex and population; mean ± s.d.)(electronic supplementary material, table S1). On average, filtered samples yielded 23 867 (18 673, 29 061; mean and CI) reads. Average alpha diversity was 0.59 (0.56, 0.61; Simpson index $(1 - D)$; electronic supplementary material, figure S1).

### (a) Microbiomes of virgin bedbugs

#### (i) Compositional differences between cuticular and genital microbiomes

Virgin bedbugs harboured distinct cuticular, external and internal genital microbiomes (PERMANOVA: Bray–Curtis: $F_{2,245} = 3.451$, $R^2 = 0.028$, $p = 0.001$; UniFrac: $F_{2,245} = 5.915$, $R^2 = 0.046$, $p = 0.001$). Multiple comparisons showed that the composition differed between the cuticle and the internal reproductive organs of both sexes (pairwise PERMANOVA with Benjamini–Hochberg correction: Bray–Curtis: $F_{1,219} = 5.264$, $R^2 = 0.001$, $q =$

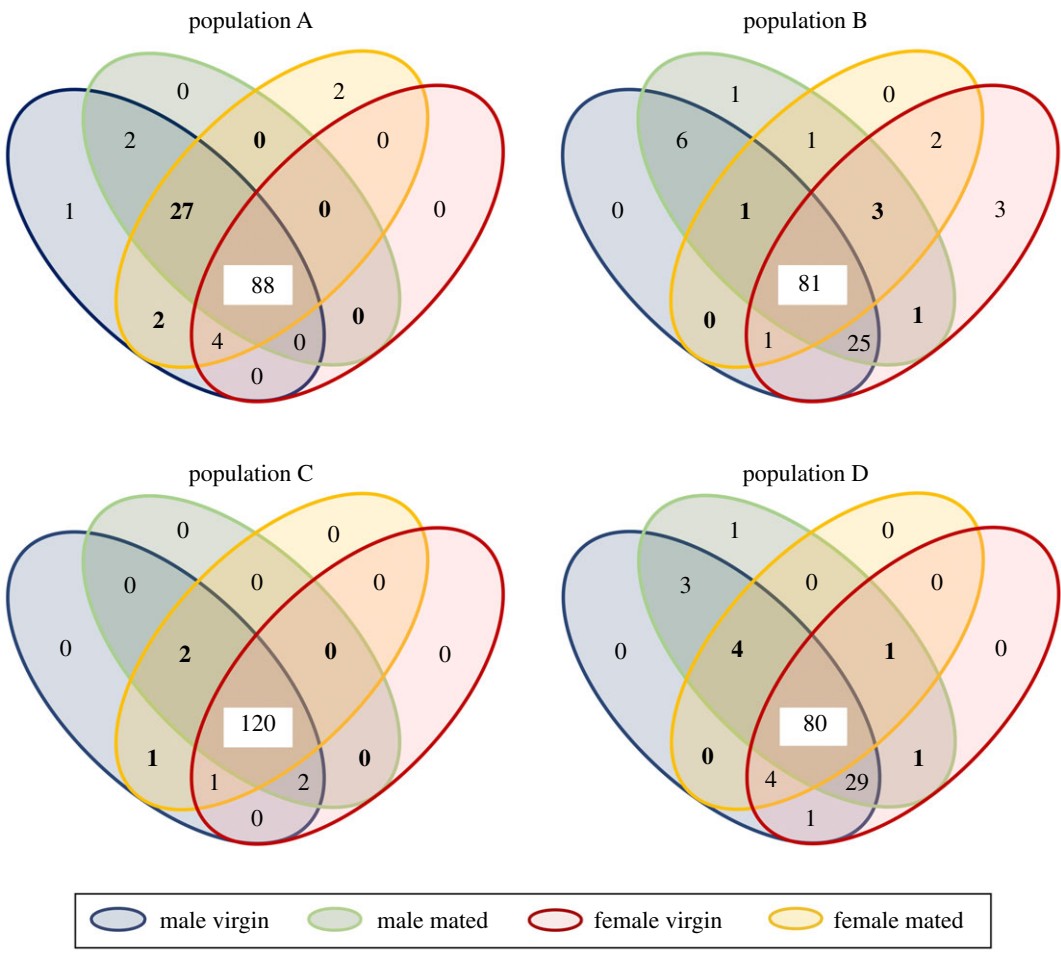

**Figure 2.** Occurrence of all SVs within the genital microbiomes split up by population, sex and mating status. Numbers on white background represent strains shared by both sexes and mating status, bold numbers strains that might have been sexually transmitted. The dataset consisted of 126 SVs, of which all were harboured by population A–C, and 124 by population D. (Online version in colour.)

0.003) and between the paramere and the internal reproductive organs of both sexes ($F_{1,178} = 2.351$, $R^2 = 0.013$, $q = 0.008$) but not between the cuticle and the paramere ($F_{1,92} = 1.404$, $R^2 = 0.118$, $q = 0.12$) (electronic supplementary material, figure S2). Cuticle, external and internal reproductive organs did not differ in between-individual difference (Multivariate test for homogeneity of variances: Bray–Curtis: $F_{2,243} = 1.467$, $p = 0.23$; UniFrac: $F_{2,243} = 0.887$, $p = 0.41$).

The compositional differences between microbiomes seem to be based on whether organs are internal or external. Environmental bacteria have been found on the external reproductive organ of bedbugs [30], suggesting cuticle and paramere are mostly colonized by environmental bacteria. By contrast, internal organs might be better protected from environmental bacteria and, therefore, are composed of a different set of bacteria. While it makes sense for internal organs to harbour symbionts that play a role in reproduction, the function of the paramere as a sperm-delivering organ is unlikely to require symbionts.

## (ii) Composition of genital microbiomes

The genital microbiomes of virgin bedbugs harboured on average 20 (15, 25; mean and CI) SVs (female) or 30 (22, 37) SVs (male). The composition of the microbial communities differed between populations (PERMANOVA: Bray–Curtis: $F_{3,170} = 1.891$, $R^2 = 0.031$, $p = 0.007$; UniFrac: $F_{3,170} = 2.307$, $R^2 = 0.036$, $p = 0.008$) (figure 1a) and sexes (Bray–Curtis: $F_{1,170} = 3.780$, $R^2 = 0.020$, $p = 0.001$; UniFrac: $F_{1,170} = 7.248$,

$R^2 = 0.038$, $p = 0.001$) (figure 1b). Even different organs from the same sex harboured distinct microbiomes (Bray–Curtis: $F_{4,170} = 1.579$, $R^2 = 0.034$, $p = 0.005$) but not when correcting for phylogeny (UniFrac: $F_{4,170} = 1.438$, $R^2 = 0.030$, $p = 0.110$) (figure 1c). Between-individual variation did not differ between populations (Multivariate test for homogeneity of variances: Bray–Curtis: $F_{3,175} = 1.420$, $p = 0.24$; UniFrac: $F_{3,175} = 1.664$, $p = 0.18$), sexes (Bray–Curtis: $F_{1,177} = 2.073$, $p = 0.15$) or organs (Bray–Curtis: $F_{5,173} = 1.462$, $p = 0.20$; UniFrac: $F_{5,173} = 1.350$, $p = 0.25$). Only when correcting for phylogeny, between-individual variation differed between sexes (UniFrac: $F_{1,177} = 4.793$, $p = 0.03$).

We did not find any SV present in all samples of a given sex, but the three most prevalent SVs occurred in at least half of all individuals within each sex. Males and females did not differ in the prevalence of a gammaproteobacterial endosymbiont of *C. lectularius* (males: 67%, females: 64%) and one *Rickettsia* strain (males: 58%, females: 59%), whereas more females than males harboured a second *Rickettsia* strain (males: 59%, females: 69%). The relative abundance of these bacteria varied tremendously from 0% to 50% (gammaproteobacterial endosymbiont) or from 0% to 100% (both *Rickettsia* strains) in individual samples. Virgin males and females shared 92 (population A), 108 (population B), 123 (population C) or 114 (population D) out of all SVs found in virgin bedbugs from the specific population (population A–C: 126, population D: 124) (figure 2).

The variation in genital microbiomes between populations indicates local adaptation of microbes and hosts and

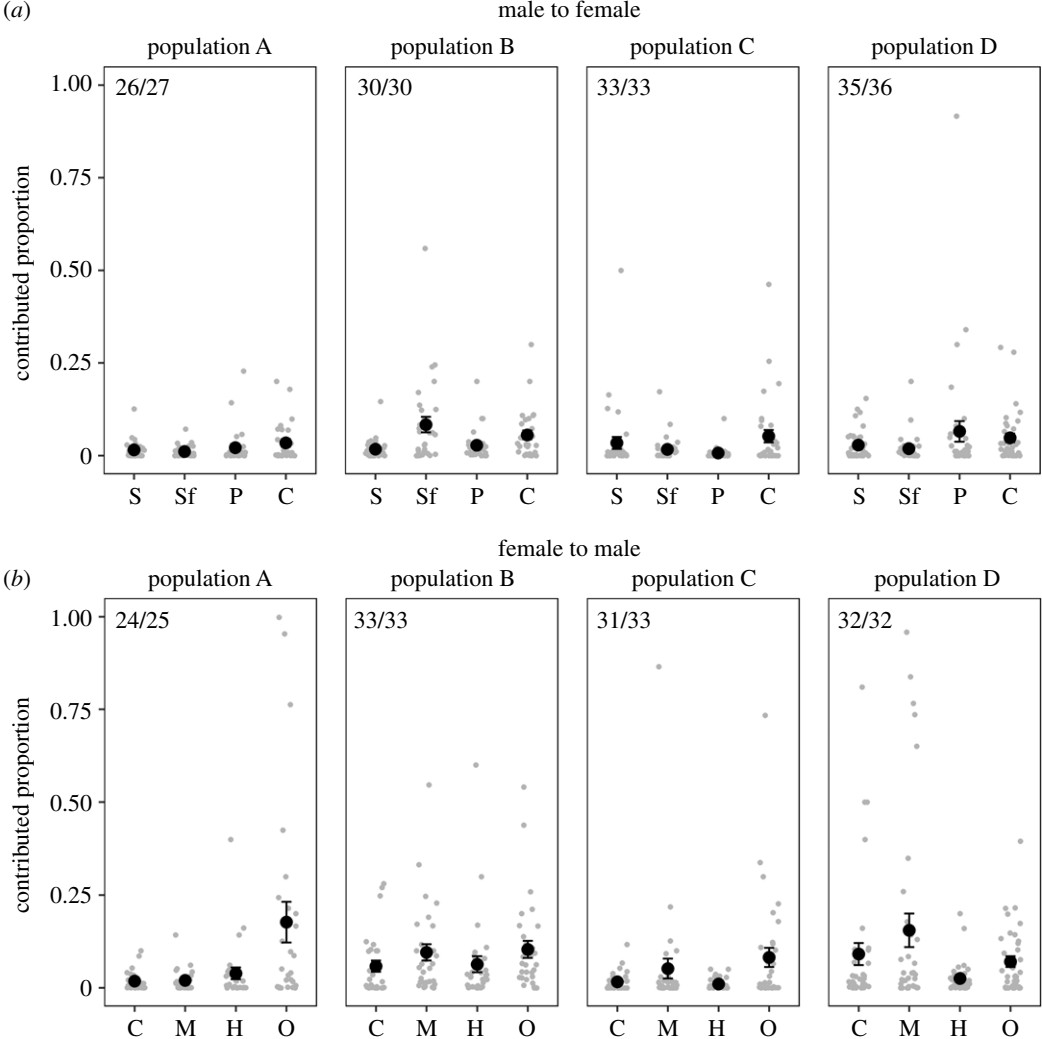

**Figure 3.** Potential for sexual transmission as estimated by SourceTracker2 [66]. Given are the proportions (*a*) that the sperm vesicle (S), the seminal fluid vesicle (Sf), the paramere (P) and the cuticle (C) of virgin males contribute to the microbiomes of mated females, and (*b*) that the cuticle (C), the mesospermalege (S), the haemolymph (H), and the ovaries (O) of virgin females contribute to the microbiomes of mated males. Depicted are means (black), standard errors of the mean, and individual data points (grey). The frequency of potential transmission for each population and transmission direction is given in the top left corner.

likely arises from host–microbe coevolution. While sexual conflict and genetic drift might affect this host–microbe interaction as well, we suspect bacteria to play an important role in shaping the evolution of reproductive traits and causing them to vary between populations. Ultimately, the coadaptation of genital microbiomes and reproductive traits might give rise to reproductive barriers between populations, leading to reproductive isolation and giving reproductive ecology an important role in speciation.

Despite of the controlled environmental factors in the laboratory, populations differed in their microbiome composition. Previous research has shown that bedbugs from infestations within the same city exhibit extreme levels of genetic differentiation [69] and that microbial communities of whole body homogenates are infestation specific [70]. In humans, vaginal microbiomes are ethnicity dependent [7], suggesting genital microbiomes might be adapted to host genotypes. In the laboratory setting, environmental bacteria are population specific in bedbugs [12]. These microbes probably originate from the faeces that are constantly deposited on the filter paper in the housing containers. Some of these bacteria might transfer to the paramere and the cuticle of both sexes and might finally be added to the genital microbiomes. Whether these differences in initial microbial communities are based on a founding effect

or whether microbe colonization is host genotype specific remains unknown. Experimental evolution experiments combined with microbe exposure treatments could show whether genital microbiomes adapt to environmental microbes and/or whether the host genotype selects for adaptations.

Whole body homogenates or gut samples in a variety of species show sex-specific microbiome compositions [14–16]. These differences could be explained by different behaviours, feeding strategies, functions in the ecosystem or roles in reproduction. Despite a lack of studies investigating the origin of sexual dimorphism in the microbiome, pronounced differences exist between the genital microbiomes of female and male red-winged blackbirds [3] and bedbugs [12,13].

To our knowledge, we provide evidence for organ-specific genital microbiomes in female insects for the first time, a finding similar to the varying microbiome composition along the female reproductive tract in humans [71], which could be caused by the different accessibility for OM or the function of the organs. However, when correcting for phylogeny, genital microbiomes were not organ specific, suggesting that bacterial strains in different organs were related, possibly because the microbiomes of all organs within one sex originate from the same initial bacterial community.

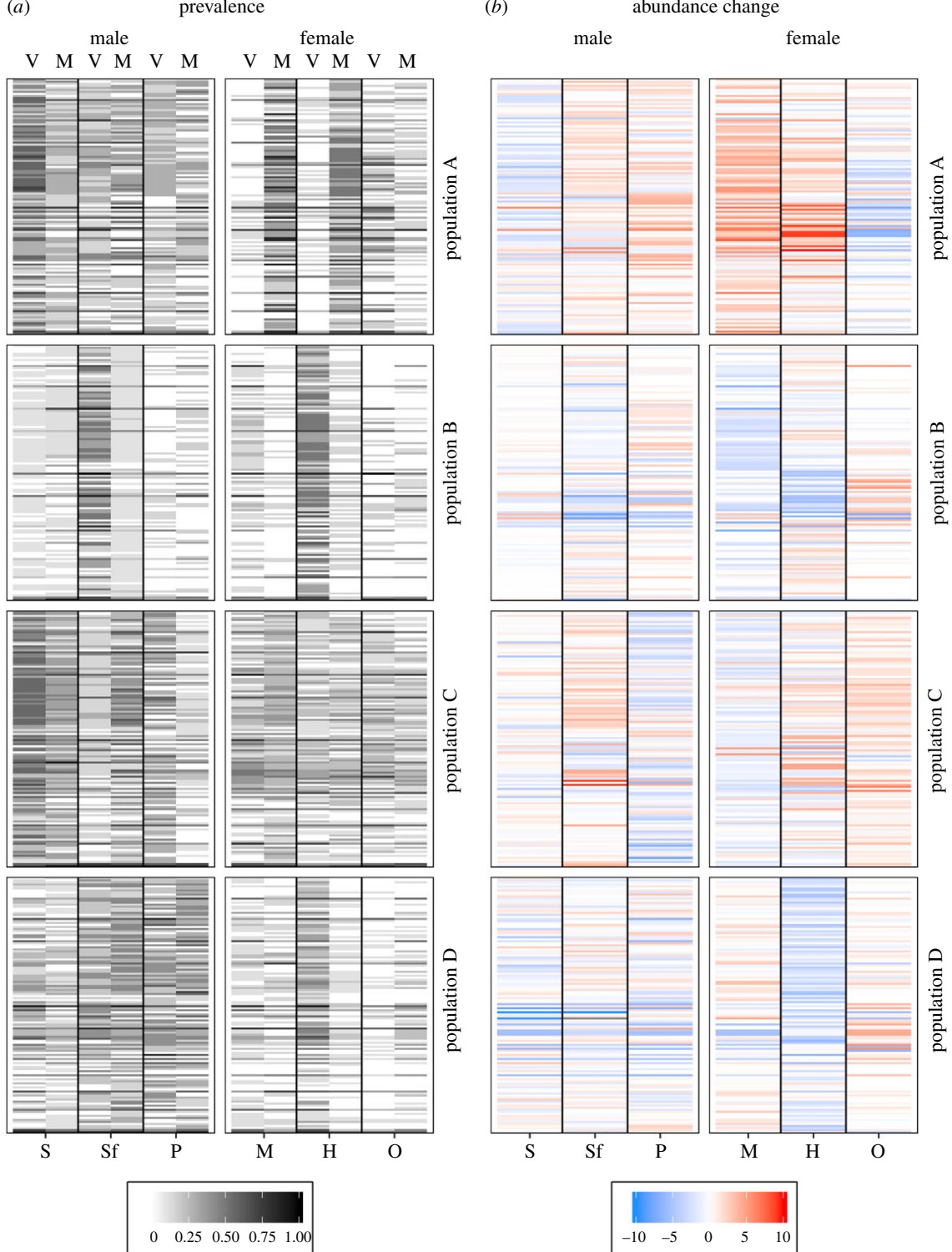

**Figure 4.** Changes in prevalence and abundance of SVs present in the microbiomes of the sperm vesicle (S), the seminal fluid vesicle (Sf), the paramere (P), the mesospermalege (M), the haemolymph (H) and the ovary (O) from virgin versus mated individuals. Given is (*a*) the prevalence of each SV before and after mating and (*b*) the log2-fold abundance change for each SV due to mating as estimated by GLM fits in the *edgeR* package [60,61]. (Online version in colour.)

## (b) Mating-induced changes in genital microbiomes

### (i) Structural changes

We found mating-induced changes in the genital microbiomes of females and males as virgin and mated individuals harboured distinct genital microbiomes (PERMANOVA: $F_{1,355} = 1.932$, $R^2 = 0.005$, $p = 0.04$) (figure 1*d*). We found no interaction of organ and mating status ($F_{5,317} = 1.003$, $R^2 = 0.013$, $p = 0.47$), population and mating status ($F_{3,317} = 0.762$, $R^2 = 0.006$, $p = 0.84$), or population, organ and mating status ($F_{15,317} = 0.777$, $R^2 = 0.031$, $p = 0.99$). Between-individual variation was similar

between mating status (Multivariate test for homogeneity of variances: $F_{1,363} = 0.010$, $p = 0.92$), organs ($F_{5,359} = 1.440$, $p = 0.21$) and populations ($F_{3,361} = 0.280$, $p = 0.84$).

In all four populations, virgin males and mated females shared bacterial strains that were not present in virgin females (figure 2). In two populations (population B and D), mated males harboured bacteria that were present in virgin females but not in virgin males (figure 2). Out of the 32 bacteria that were potentially transmitted from males to females, one strain (unclassified Comamonadaceae) was shared by population A and B, two strains (both *Pseudomonas* sp.) by

population A and C, and another two strains (unclassified Lactobacillaceae and *Cloacibacterium* sp.) by population A and D (for a detailed list of transmitted bacteria species see electronic supplementary material, table S6). None of the bacterial strains were potentially transmitted from females to males in more than one population (for a detailed list of transmitted bacteria species see electronic supplementary material, table S7). All bacteria that were only found in the genital microbiomes of mated individuals occurred on the cuticle of virgin bedbugs. According to SourceTracker2, bacteria in mated individuals likely originated from at least one of the organs of the opposite sex in 96% to 100% (mated females) (figure 3a) and 94% to 100% (mated males) (figure 3b) of the cases.

In accordance with the previous studies in vertebrates [23,24,72,73] and invertebrates [12,13], we found mating-induced changes in the genital microbiomes of bedbugs. Changes in strain composition can be caused by immunological substances targeting members of the genital microbiome, by the sexual transmission of bacteria via the ejaculate or male genitalia, by strains transferred from the cuticle or by strains invading genital openings and copulatory wounds. Indeed, our results indicate that a part of the bacteria in mated individuals of one sex originate from the reproductive organs of the opposite sex as well as from the cuticle. Bacteria from the cuticle invading the genital microbiomes are highly likely to be OM rather than bacteria causing STI, which is in accordance with a previous study in bedbugs using culture-based methods [30].

Surprisingly, even males seemed to be subject to transmission, although males are less likely to face copulatory wounding, and the distance between the environment and the internal reproductive organs is larger compared with females. However, in case males do not apply pressure to transfer their ejaculate, bacteria might reach the internal organs through the ejaculatory duct via a capillary effect similar to the invasion of human testicles by urethral pathogens [74]. Antimicrobial peptide production in the genital tract of *Drosophila* males in response to bacteria deposited on the genital plate [75] suggests microbes regularly enter the genital microbiomes of male insects. Experimental manipulation of bacteria on the paramere might clarify whether and how bacteria can enter the paramere and move through the ejaculatory duct towards the internal organs.

## (ii) Prevalence and abundance changes

The prevalence of many bacterial strains was changed by mating but there was no clear direction of change, i.e. decrease or increase in prevalence. However, the prevalence of several bacteria seemed to change simultaneously within the same organ, sex and population (figure 4a). Mating affected the abundance of several bacterial strains in all organs except for sperm vesicle, paramere and haemolymph samples from population B and mesospermalege samples from population D. The proportion of differentially abundant SVs did not differ between males and females ($F_{1,19} = 3.254$, $p = 0.09$) or between populations ($F_{3,19} = 1.56$, $p = 0.23$) and population did not interact with sex ($F_{3,16} = 0.367$, $p = 0.78$). No clear

direction of abundance change for populations, sexes or organs was identified (figure 4b).

Genital microbiomes of females should be more affected by invading bacteria because bacteria could enter the immune organ via the ejaculate and via copulatory wounds. In accordance with this idea, sexual transmission of bacteria in birds is higher when males are the transmitting sex [72]. Since mating induced prevalence and abundance changes of several bacteria in both sexes of the common bedbug, even the microbiomes of males seem to be affected by mating. Bacteria can decrease survival [76] but also cause a trade-off between immunity and mating and hence decrease fecundity [77]. Moreover, bacteria could harm sperm within the male and female directly. OM decrease sperm motility [33–37] and incapacitate spermatozoa [36] in humans, at least *in vitro*, and environmental bacteria increase sperm mortality in bedbugs [38]. To reduce the costs of mating-associated infections, bedbugs have evolved the mesospermalege [76]. The high number of haemocytes [46] able to phagocytose bacteria [78] in this organ might stabilize its microbiome and protect against invading bacteria. In addition, lysozyme in the seminal fluid of males [79] and in the mesospermalege produced in anticipation of mating [80] could help to reduce invading bacteria. Furthermore, endosymbionts have been shown to interact with invading microbes [42–45] and might help to control non-resident bacteria in the genital microbiomes. Future studies should investigate the effect of the species with the largest prevalence and abundance changes on fecundity and survival and what adaptations have evolved to eliminate the possible threat to host integrity.

We have demonstrated that genital microbiomes of the common bedbug *C. lectularius* differ between populations, sexes and organs. Our findings show that genital microbiomes are sensitive to mating, an activity that every sexually reproducing animal experiences. Future studies should investigate sexual transmission dynamics of OM in combination with fitness effects on both sexes. Experimental manipulation of the female immune system could provide information about the importance of immunity in response to genitalia-associated bacteria. Finally, the coadaptation of genital microbiomes and reproductive traits might lead to reproductive isolation between populations, giving reproductive ecology an important role in speciation.

Data accessibility. Sequencing reads were deposited in NCBI's Sequence Read Archive with the accession number PRJNA560165.

Authors' contributions. O.O., P.R.J. and S.B. conceived the idea and designed the experiment. S.B. and S.M. carried out the experiment. S.B. and P.R.J. performed the bioinformatics and statistical analysis. S.B., P.R.J., S.M. and O.O. interpreted the results and S.B. and O.O. wrote the manuscript. All authors read and approved of the final manuscript.

Competing interests. The authors declare no competing interests.

Funding. S.B. was supported by a grant of O.O. awarded by the German Research Foundation (grant no. OT 521/2-1).

Acknowledgements. We would like to thank M. Kaltenpoth and two anonymous reviewers for helpful comments on the manuscript.

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
