## [Reviewer comments · Proceedings of the Royal Society B: Biological Sciences]

Review History

RSPB-2019-2567.R0 (Original submission)

Review form: Reviewer 1

Recommendation

Major revision is needed (please make suggestions in comments)

Scientific importance: Is the manuscript an original and important contribution to its field?

Excellent

General interest: Is the paper of sufficient general interest?

Good

Quality of the paper: Is the overall quality of the paper suitable?

Acceptable

Is the length of the paper justified?

Yes

Should the paper be seen by a specialist statistical reviewer?

No

Do you have any concerns about statistical analyses in this paper? If so, please specify them explicitly in your report.

Yes

It is a condition of publication that authors make their supporting data, code and materials available - either as supplementary material or hosted in an external repository. Please rate, if applicable, the supporting data on the following criteria.

Is it accessible?

No

Is it clear?

Yes

Is it adequate?

N/A

Do you have any ethical concerns with this paper?

No

Comments to the Author

A recent explosion of studies are demonstrating the importance of the microbiome for host biology. These studies, however, are primarily focused on studies of the gut microbiome, and the majority are on humans or model lab species. In contrast, microbial communities associated with reproductive organs remain relatively unexplored and are rarely considered from an ecological or evolutionary perspective. This is despite the potentially important role these communities may play in reproduction and reproductive processes.

This manuscript presents the results of a study investigating the microbiome of male and female genitalia in the common bedbug, and analyses aimed at testing for differences in microbiomes among populations, between the sexes, and in different reproductive organs, as well as the impact of mating on these genital microbiomes. The authors report distinct microbiomes in the different reproductive organs/cuticle of virgin males and females, though alpha-diversity levels seem not to vary considerably among the sexes, populations, or between mated/virgin individuals. In my opinion, the two most notable results reported in the manuscript are: (1) the finding that mating changed the microbiome composition (community composition and relative abundance of specific SVs) of males and females, and (2) the finding that populations show significant differences in microbiome composition.

In the first instance, some recent studies have demonstrated changes in the microbiome following mating. More broadly, monogamous mating partners appear to share microbes through sexual transmission {Kulkarni:2007ik} and mating can lead to increased similarity in the microbiome of both males and females {Kreisinger:2015bv, White:2010wa, Mandar:2015eu} as well as alterations in these microbial communities {Schoenmakers:2019ds, White:2010wa}. Studies of invertebrates are generally lacking, however, and as such this study makes a significant contribution to a small, but growing, body of literature. Nonetheless, I do think the paper would benefit from more fully placing the results of this study in the context of the current literature, and thus discussing the results with reference to the above literature both in the introduction and/or discussion text. In addition, I think a discussion whether changes are due to sexual transmission of microbes versus the disruption of the existing microbiome is warranted.

In the second instance, the finding of population differentiation in genital microbiomes is a particularly novel and interesting finding. As the authors of the manuscript discuss, this has very interesting potential ramifications for the evolution of reproductive isolation and speciation. I particularly like this aspect of the study and thus would like to see the results discussed in a little

more detail. For example, can then authors perhaps place these findings in the broader context of parasite-mediated speciation.

Overall, I really liked this study and I think it makes a significant contribution to an emerging field of study on the microbiomes of reproductive systems. However, I think the manuscript would benefit from some additional information and, in some instances, greater clarity in the written text. To facilitate a revision of the manuscript, I have provided a number of comments below (noted by line number) to highlight my concerns or identify text that is unclear.

Line 26: there is a word missing before the text 'specific organ systems' perhaps it should be 'in specific organ systems'

Line 27: I suggest this should be e.g. genital microbiomes, as there are other reproductive organs that harbour microbiomes.

Line 39: delete 'again'

Line 49: it may be more appropriate to say gastrointestinal microbiome instead of gut in this sentence as the gastrointestinal system is more technically regarded as an organ system.

Line 53: The term intracellular reproductive manipulator is an unfamiliar one. Could the authors use a more familiar term, or perhaps provide an example of an intracellular reproductive manipulator in order to help the reader think about this information.

Line 60/61: I would say the composition of these microbiomes are generally unknown for most species, vertebrates and invertebrates alike. I suggest the authors note this and highlight that this is especially the case for insects.

Line 65: I think this section could be reworded somewhat. In the sentence ending on line 65 there is mention of the vaginal microbiome, and yet the next sentence talks about genital microbiomes. If the vaginal microbiome is not a genital microbiome, this needs to be made clear in the definition of the genital microbiome. However, the text suggests they are considered genital microbiomes as the example then given (on line 65/66) concerns the vaginal microbiome.

Line 70: see comment above regarding previous studies demonstrating shifts in microbiome composition with mating.

Line 72: can the authors please clarify, what defines an opportunistic microbe (OM)? Is this just a microbe from an environmental source such that the microbiome is a random assemblage of species? Or is it somehow different from a microbe from the environment. The terms seem to be used interchangeably in the manuscript, and as a result it is unclear why this term is used or if it is somehow a subset of microbes that might be from an environmental source. I think a little clarity in what an OM is would be useful for the reader. I think it might also be worth considering if an OM is from an environmental source whether all microbes have an equal opportunity to be an OM or whether there are some processes that influence whether or not an environmental microbe invades or colonises the microbiome of an organ, such as the genitalia.

Line 75: why might OM have a more direct effect compared to STM?

Line 82: It is not clear to me why the example of intestinal microbiome of mice is used as an example of sexually transmitted microbes disturbing the resident microbiome.

Line 94: I think ethnicity is a more correct term than ancestry. In the original publication it is mentioned as ethnicity. I think this supports the idea of population level variation in microbiomes more clearly.

Line 116/117: please provide a citation for the sentence that microbial communities differ between organs and mating increases microbiome similarity between males and females. More generally, I think the authors previous work could be explained in more detail in order to clarify the aims of the current study and to help distinguish between the findings of the previous publication and this current paper. I think this will more clearly set out what is known in the system and what this paper adds to our knowledge.

Line 123/124: is it differences in external environment or population level variation that is being investigated. I thought the latter.

Line 125: by groups, do you mean populations? Please clarify.

Line 136: Given that results are presented first, I think it needs to be made clear what reproductive organs are being investigated. More generally, in the text it is frequently unclear what an internal versus external organ is. I think the authors could present the information on what organs were investigated and whether these are internal or external in a more clear and explicit way.

Line 143: differences between cuticle and genital microbiomes are not always significant, thus I think the first sentence of this results section is not true. Perhaps this text could be reworded.

Line 143-148: given that the results are presented first, I think it is important to tell the reader how microbiomes were assessed. For example, 16S rRNA amplicon sequencing was used to investigate the structure of the microbial communities.

Line 145-147: q instead of p in p-values.

Line 144-147: the text states '...their relative abundance....but not between the cuticle and the external..... or the internal reproductive organs of both sexes and the paramere....' but the p values stated is 0.02, which suggests a significant difference. Thus, either the text or the statistics reported seem to be in error.

Line 147: what 'groups'?

Line 148: how was between-individual difference estimated? I know this is in the methods, but not including it here means the reader has to flip back to the methods to understand the results. Thus, I think the authors should makes these measures clear in the results section.

Line 150: by 'location' do you mean site on the host? Or population location?

Line 156: by 'other organs' do the authors mean all other organs investigated. At this point I had kind of lost track of the different organs being investigated, so I think it would help the reader to be explicit here.

Line 157: can the authors clarify what they mean by the function of the organ driving the finding?

Line 161: which genital microbiomes? All? Just males? Females? If they are all lumped, is this appropriate?

Line 162: again, what is meant by 'group' here.

Line 164: Is the comparison of the sexes mentioned here for all organs investigated lumped together?

Line 164: I advise the use of panels (A-D) on Figure S3 to clarify to the reader which graph to look at.

Line 165: Should the results of different organs from the same sex also point to figure S3?

Line 169: Are these then the most prevalent of the SVs identified in the study?

Line 172: any ideas for why this Rickettsia strain was more prevalent in females?

Line 174-176: I think it would be useful to also show the actual number of shared SVs relative to the total number of SVs (e.g. 44/50) in addition to the % as the information in Figure 1 extends beyond that of the differences between virgin males and females.

Line 182-189: I greatly appreciate that the authors do not want to speculate too widely about their findings. However, I think it would be useful here to discuss species and population-specific microbiomes more broadly (i.e. discuss microbiomes of a range of host sites). For example, a number of studies have shown variation in microbiomes in relation to host phylogeny.

Line 204: the sub heading here mentioned relative abundance of SV, but it is a little confusing with the subheading on line 248 which deals with abundance changes. Perhaps the authors could change the wording to make the headings clearer and the difference between them more explicit.

Line 206: As before, I think it is necessary to indicate how these data were generated and how specific metrics estimated.

Line 207: which part of figure S3 is referenced here? Panels would be useful.

Line 213: Which two out of four populations?

Line 216: replace 'q' with 'p'

Line 228: Not being familiar with this method, I found the results/discussion and the methods section on the analysis of nestedness and turnover very difficult to follow. I suggest that the authors explain this methodology in greater detail, including some mention of why these processes are expected to account for changes in the microbiome. It is difficult to understand exactly how strain replacement differs from strain loss/introduction.

Line 249: I think the authors might consider more graphical ways in which to display their data. For example, I think the change in microbiome composition (what is present and the abundance of the different SVs) could be more clearly conveyed using a heatmap approach.

Line 257-258: I think this is a very interesting result (ie. that abundance change did not vary by sex). However, I think the idea that females are expected to be more affected by mating needs to be more clearly laid out and supported by current literature.

Line 260: Can bacteria more readily enter the female body? Your results suggest this is perhaps not the case in bedbugs? I suggest the authors add a citation to support this idea.

Line 288-289: why are the populations ordered A, C, D and then B when presented for the first time in the methods. I suggest they should be presented in alphabetical order.

Line 287-295: A consider more amount of detail is necessary in this section of the methods. When were the wild populations caught? How long were they maintained in a captive setting before the study was performed? Where were the populations maintained? All in the same location? If so, how might that change the expectations of environmental microbes invading the genital

microbiome. Would this standardised housing not homogenise the among-population differences? If so, then I think the paper needs to pay closer consideration of the potential genetic versus environmental impacts of microbes and what factors determine the presence of particular members of a microbiome. How long were they held in a standard environment? Did they all have the same bedding? Was it sterilised before use? Did they all have the same food source? Was it sterilised? I think all of these issues need to be clear in the manuscript. Furthermore, if these conditions were all standard, I think the authors need to think carefully about what they consider an environmental microbiome versus population differences.

Line 304-306: I greatly appreciate the authors methods to prevent contamination, this was very pleasing to see.

Line 313: see general comment below regarding what the genital microbiome is.

Line 317-318: Again, I greatly appreciate the use of sampling controls at this stage in the study to deal with the issue of contamination. This is very pleasing to see in a microbiome study.

Line 321-325: Please provide more details of the DNA extraction approach. Did you follow the kit protocol? It is also unclear if you used a bead beating step. If you did, please make this clear and provide full details of the methods used.

Line 325: Again, great to see the use of controls! Also the use of PCR controls!

Line 329: how many controls run in each library/plate?

Line 341-345: any use of PhiX? Was it 250 bp paired end reads?

Line 359-361: I am very much in support of this approach to deal with contamination. However, I would like to see a bit more information about the contaminants in the supplementary material. What was there? How frequently was it a problem in the controls? Did it affect all controls (e.g. PCR controls, sampling controls, etc.)?

Line 364: It is not clear to me why environmental samples were excluded. What do you mean, any bacteria from environmental sources? Similarly, why exclude uncultured samples? Do you mean these were excluded from the database used for taxonomic assignment? This needs to be made clearer.

Line 381: Which are the internal versus the external reproductive organs? I think this needs to be clearly stated at some point.

Line 398: did you do any model comparison testing to see if the removal of non-significant interactions was a significant improvement to the model.

Line 410: why not use Manhattan distances and PCoA for this analysis instead of creating a presence/absence matrix?

Line 405-417: Given that the analyses presented in this section are presence/absence data I wonder if there could be an issue with 0 inflated data? Did the authors consider this statistical issue in their analysis? Could overdispersion be an issue? Perhaps the authors can comment on this and/or show that this isn't a problem for their data and models.

Line 663: I would recommend trying to make the symbols differentiating the populations more obvious. Currently it is difficult to see the spread of the data very clearly. This comment is also relevant to Figure S1.

Line 696: I think it would be better to have a figure showing the differences between populations (such as the top, left panel of Fig S3) in the main text.

General comments:

1. Defining the genital microbiome. I think the manuscript would benefit from a clearer explanation of what is and what isn't a genital microbiome. For example, I was often unsure if the genital microbiome simply referred to external structures, such as a paramere of males or if this term also included the sperm vesicles.
2. Figure 1 shows a number of different contrasts, and as a result it often becomes difficult to extract the information pointed to in the text (e.g. see comment on line 174-176). I think the authors might consider parsing out the information in Figure 1 into multiple figures.
3. Supplementary tables should have a legend to make sure they can be understood without referring to the main text. Currently they are only briefly mentioned in the main text and this makes it difficult for the reader to follow the flow of information.
4. I wonder if the authors might consider using phylogeny-based methods, such as UniFrac (weighted and/or unweighted), to investigate microbiome composition (presence/absence and abundance). These metrics might show different results that would be well worth knowing, or add further support to the findings of the study.
5. I think it would be beneficial to show the population differences using a PCoA or NMDS plots

Review form: Reviewer 2

Recommendation

Major revision is needed (please make suggestions in comments)

Scientific importance: Is the manuscript an original and important contribution to its field?

Good

General interest: Is the paper of sufficient general interest?

Acceptable

Quality of the paper: Is the overall quality of the paper suitable?

Good

Is the length of the paper justified?

No

Should the paper be seen by a specialist statistical reviewer?

Yes

Do you have any concerns about statistical analyses in this paper? If so, please specify them explicitly in your report.

Yes

It is a condition of publication that authors make their supporting data, code and materials available - either as supplementary material or hosted in an external repository. Please rate, if applicable, the supporting data on the following criteria.

Is it accessible?

No

Is it clear?

Yes

Is it adequate?

Yes

Do you have any ethical concerns with this paper?

No

Comments to the Author

In this manuscript, Bellinva and colleagues report changes in microbial communities in the genital microbiomes of *Cimex lectularius* as a result of mating. The authors report several interesting findings: first, the microbiomes of lab grown *C. lectularius* populations from different geographic locations and collected in different years differ. Second, microbiomes of males and females differ. And third, mating results in changes in genital microbiomes. The authors build on previous reports from the same group; While this is an incremental addition on top of their previous reports, validating the effect of mating on both cuticular and internal microbiomes in different populations collected over different years and locations is of interest. I raise several questions and comments, and suggest a few additional analyses or experiments that would strengthen this manuscript before it is ready for publication.

Introduction:

Generally speaking, the Introduction is very long (~30% of the main text) and often speculative. Speculations and interpretations should be part of the discussion, rather than introduction, and often there is redundancy between the text in the introduction and discussion.

At times the authors make the distinction between sexually transmitted microbes and sexually transmitted opportunistic microbes. While this might be useful in trying to distinguish between pathogens that are sexually transmitted and microbes that happen to be transferred but aren't pathogens, it is confusing and unclear in the text.

Methods: In general, the authors have made an effort to properly control and analyze their data reliably.

Different cultures originated from different geographic locations and were maintained in the lab for different times (9 to over 20 years). How do these populations differ genetically? As the authors note, diet and growth conditions are major drivers in shaping the microbiome and the differences between populations might suggest that a founder effect where initial microbial communities differed significantly, or, alternatively, that genetic differences between these populations affected colonization. Both of these would be of interest, or at least acknowledged. How successful was mating over the 60s given to bugs? Do all mate? Was mating done in a sterile chamber? Were males added before females or vice versa?

Bioinformatic and statistical analyses

How many reads were obtained per sample, or per group on average? Were data subsampled (for beta diversity) or samples with less than a certain number of reads omitted?

The methods report 147 SVs, while Table S3 has 165 SVs.

Nestedness and turnover are interesting metrics. Would also be interesting to use SourceTracker (Knights et al 2011, <https://github.com/biota/sourcetracker2>) to test whether SVs from virgins in each population are source for specific SVs in mated populations.

Why was FDR 1% used for EdgeR?

Results and Discussion:

While 643 samples were sequenced, only 495 samples were analyzed. It is unclear what were the filtering criteria beyond singletons, 0.01% of total reads (how many reads did you collect in total?) and how they affected the data that was left. Generally speaking, losing ~25% of your samples to QC filtration might suggest that the data is not of high quality and that there is an issue with your sequencing method or instrumentation.

This might be a misunderstanding on my part, and I apologize if it is - some of the F statistics reported seems rather low for the reported p values (or q-values). For example $F_{1,177}=2.350$, $q = 0.02$ (which by an F calculator would be 0.13), $F_{3,170}=1.885$, $p = 0.004$ (calculated to be $p=0.12$). In addition, this difference is reported as not significant while <0.05

A direct comparison between populations in virgin bugs would be beneficial in setting up the story. It is mentioned (lines 149-150) but not presented until later. Related to this - the authors should discuss what could be driving microbiome differences in these lab grown populations. Could the experimental setup for mating facilitate bacterial transfer through the media or environment?

It would be useful to report what microbes were found to be transferred by mating, rather than just the numbers of SVs. moreover, it would be useful to compare if the same Genera (or other taxonomic level you are interesting in) are transmitted by mating across populations. At the moment, you are only addressing it by numbers and it is not very informative.

In addition, reporting that few SVs were different between mated and unmated doesn't say much about their relative abundance and how well they were represented.

Decision letter (RSPB-2019-2567.R0)

17-Dec-2019

Dear Ms Bellinva:

I am writing to inform you that your manuscript RSPB-2019-2567 entitled "Mating changes the genital microbiome in both sexes of the common bedbug *Cimex lectularius* across populations" has, in its current form, been rejected for publication in *Proceedings B*.

This action has been taken on the advice of referees, who have provided two very thorough reviews. We are all agreed that this is a potentially really valuable study, but the referees have indicated that substantial revisions are necessary. With this in mind we would be happy to consider a resubmission, provided the comments of the referees are fully addressed. However please note that this is not a provisional acceptance.

- 1) A 'response to referees' document including details of how you have responded to the comments, and the adjustments you have made.
- 2) A clean copy of the manuscript and one with 'tracked changes' indicating your 'response to referees' comments document.

3) Line numbers in your main document.

Yours sincerely,
 Professor Loeske Kruuk
 Editor
 mailto: proceedingsb@royalsociety.org

Associate Editor
 Board Member: 1

Comments to Author:

This manuscript investigates the genital microbiome in a traumatically inseminating insect, and how it changes as a result of mating. The authors find evidence that some species of microbes are indeed transmitted between partners when mating, and there also seemed to be some consistent differences between sexes and populations. Overall the results are rather descriptive, but this type of study is still quite novel, especially in insects. The speculation at the end of the discussion about how genital microbiomes could be related to speciation and the evolution of reproductive isolation is particularly interesting. This has the potential to be an important and previously unstudied mechanism.

Both reviewers were generally positive towards the manuscript, but both also had rather extensive suggestions for improvement, especially reviewer 1. Most of the suggestions are to do with clarifying various aspects of the methods and should be relatively straightforward to address. However some of the suggestions involve additional statistical analysis. Because of this, I am recommending rejection with possibility to resubmit, in case the additional analyses will result in a different interpretation of the results.

Reviewer(s)' Comments to Author:

Referee: 1

Comments to the Author(s)

A recent explosion of studies are demonstrating the importance of the microbiome for host biology. These studies, however, are primarily focused on studies of the gut microbiome, and the majority are on humans or model lab species. In contrast, microbial communities associated with reproductive organs remain relatively unexplored and are rarely considered from an ecological or evolutionary perspective. This is despite the potentially important role these communities may play in reproduction and reproductive processes.

This manuscript presents the results of a study investigating the microbiome of male and female genitalia in the common bedbug, and analyses aimed at testing for differences in microbiomes among populations, between the sexes, and in different reproductive organs, as well as the impact of mating on these genital microbiomes. The authors report distinct microbiomes in the different reproductive organs/cuticle of virgin males and females, though alpha-diversity levels seem not to vary considerably among the sexes, populations, or between mated/virgin individuals. In my opinion, the two most notable results reported in the manuscript are: (1) the finding that mating changed the microbiome composition (community composition and relative abundance of specific SVs) of males and females, and (2) the finding that populations show significant differences in microbiome composition.

In the first instance, some recent studies have demonstrated changes in the microbiome following mating. More broadly, monogamous mating partners appear to share microbes through sexual transmission {Kulkarni:2007ik} and mating can lead to increased similarity in the microbiome of both males and females {Kreisinger:2015bv, White:2010wa, Mandar:2015eu} as well as alterations in these microbial communities {Schoenmakers:2019ds, White:2010wa}. Studies of invertebrates are generally lacking, however, and as such this study makes a significant contribution to a small, but growing, body of literature. Nonetheless, I do think the paper would benefit from more fully placing the results of this study in the context of the current literature, and thus discussing the results with reference to the above literature both in the introduction and/or discussion text. In addition, I think a discussion whether changes are due to sexual transmission of microbes versus the disruption of the existing microbiome is warranted.

In the second instance, the finding of population differentiation in genital microbiomes is a particularly novel and interesting finding. As the authors of the manuscript discuss, this has very interesting potential ramifications for the evolution of reproductive isolation and speciation. I particularly like this aspect of the study and thus would like to see the results discussed in a little more detail. For example, can then authors perhaps place these findings in the broader context of parasite-mediated speciation.

Overall, I really liked this study and I think it makes a significant contribution to an emerging field of study on the microbiomes of reproductive systems. However, I think the manuscript would benefit from some additional information and, in some instances, greater clarity in the written text. To facilitate a revision of the manuscript, I have provided a number of comments below (noted by line number) to highlight my concerns or identify text that is unclear.

Line 26: there is a word missing before the text 'specific organ systems' perhaps it should be 'in specific organ systems'

Line 27: I suggest this should be e.g. genital microbiomes, as there are other reproductive organs that harbour microbiomes.

Line 39: delete 'again'

Line 49: it may be more appropriate to say gastrointestinal microbiome instead of gut in this sentence as the gastrointestinal system is more technically regarded as an organ system.

Line 53: The term intracellular reproductive manipulator is an unfamiliar one. Could the authors use a more familiar term, or perhaps provide an example of an intracellular reproductive manipulator in order to help the reader think about this information.

Line 60/61: I would say the composition of these microbiomes are generally unknown for most species, vertebrates and invertebrates alike. I suggest the authors note this and highlight that this is especially the case for insects.

Line 65: I think this section could be reworded somewhat. In the sentence ending on line 65 there is mention of the vaginal microbiome, and yet the next sentence talks about genital microbiomes. If the vaginal microbiome is not a genital microbiome, this needs to be made clear in the definition of the genital microbiome. However, the text suggests they are considered genital microbiomes as the example then given (on line 65/66) concerns the vaginal microbiome.

Line 70: see comment above regarding previous studies demonstrating shifts in microbiome composition with mating.

Line 72: can the authors please clarify, what defines an opportunistic microbe (OM)? Is this just a microbe from an environmental source such that the microbiome is a random assemblage of

species? Or is it somehow different from a microbe from the environment. The terms seem to be used interchangeably in the manuscript, and as a result it is unclear why this term is used or if it is somehow a subset of microbes that might be from an environmental source. I think a little clarity in what an OM is would be useful for the reader. I think it might also be worth considering if an OM is from an environmental source whether all microbes have an equal opportunity to be an OM or whether there are some processes that influence whether or not an environmental microbe invades or colonises the microbiome of an organ, such as the genitalia.

Line 75: why might OM have a more direct effect compared to STM?

Line 82: It is not clear to me why the example of intestinal microbiome of mice is used as an example of sexually transmitted microbes disturbing the resident microbiome.

Line 94: I think ethnicity is a more correct term than ancestry. In the original publication it is mentioned as ethnicity. I think this supports the idea of population level variation in microbiomes more clearly.

Line 116/117: please provide a citation for the sentence that microbial communities differ between organs and mating increases microbiome similarity between males and females. More generally, I think the authors previous work could be explained in more detail in order to clarify the aims of the current study and to help distinguish between the findings of the previous publication and this current paper. I think this will more clearly set out what is known in the system and what this paper adds to our knowledge.

Line 123/124: is it differences in external environment or population level variation that is being investigated. I thought the latter.

Line 125: by groups, do you mean populations? Please clarify.

Line 136: Given that results are presented first, I think it needs to be made clear what reproductive organs are being investigated. More generally, in the text it is frequently unclear what an internal versus external organ is. I think the authors could present the information on what organs were investigated and whether these are internal or external in a more clear and explicit way.

Line 143: differences between cuticle and genital microbiomes are not always significant, thus I think the first sentence of this results section is not true. Perhaps this text could be reworded.

Line 143-148: given that the results are presented first, I think it is important to tell the reader how microbiomes were assessed. For example, 16S rRNA amplicon sequencing was used to investigate the structure of the microbial communities.

Line 145-147: q instead of p in p-values.

Line 144-147: the text states '...their relative abundance.....but not between the cuticle and the external..... or the internal reproductive organs of both sexes and the paramere.....' but the p values stated is 0.02, which suggests a significant difference. Thus, either the text or the statistics reported seem to be in error.

Line 147: what 'groups'?

Line 148: how was between-individual difference estimated? I know this is in the methods, but not including it here means the reader has to flip back to the methods to understand the results. Thus, I think the authors should make these measures clear in the results section.

Line 150: by 'location' do you mean site on the host? Or population location?

Line 156: by 'other organs' do the authors mean all other organs investigated. At this point I had kind of lost track of the different organs being investigated, so I think it would help the reader to be explicit here.

Line 157: can the authors clarify what they mean by the function of the organ driving the finding?

Line 161: which genital microbiomes? All? Just males? Females? If they are all lumped, is this appropriate?

Line 162: again, what is meant by 'group' here.

Line 164: Is the comparison of the sexes mentioned here for all organs investigated lumped together?

Line 164: I advise the use of panels (A-D) on Figure S3 to clarify to the reader which graph to look at.

Line 165: Should the results of different organs from the same sex also point to figure S3?

Line 169: Are these then the most prevalent of the SVs identified in the study?

Line 172: any ideas for why this Rickettsia strain was more prevalent in females?

Line 174-176: I think it would be useful to also show the actual number of shared SVs relative to the total number of SVs (e.g. 44/50) in addition to the % as the information in Figure 1 extends beyond that of the differences between virgin males and females.

Line 182-189: I greatly appreciate that the authors do not want to speculate too widely about their findings. However, I think it would be useful here to discuss species and population-specific microbiomes more broadly (i.e. discuss microbiomes of a range of host sites). For example, a number of studies have shown variation in microbiomes in relation to host phylogeny.

Line 204: the sub heading here mentioned relative abundance of SV, but it is a little confusing with the subheading on line 248 which deals with abundance changes. Perhaps the authors could change the wording to make the headings clearer and the difference between them more explicit.

Line 206: As before, I think it is necessary to indicate how these data were generated and how specific metrics estimated.

Line 207: which part of figure S3 is referenced here? Panels would be useful.

Line 213: Which two out of four populations?

Line 216: replace 'q' with 'p'

Line 228: Not being familiar with this method, I found the results/discussion and the methods section on the analysis of nestedness and turnover very difficult to follow. I suggest that the authors explain this methodology in greater detail, including some mention of why these processes are expected to account for changes in the microbiome. It is difficult to understand exactly how strain replacement differs from strain loss/introduction.

Line 249: I think the authors might consider more graphical ways in which to display their data. For example, I think the change in microbiome composition (what is present and the abundance of the different SVs) could be more clearly conveyed using a heatmap approach.

Line 257-258: I think this is a very interesting result (ie. that abundance change did not vary by sex). However, I think the idea that females are expected to be more affected by mating needs to be more clearly laid out and supported by current literature.

Line 260: Can bacteria more readily enter the female body? Your results suggest this is perhaps not the case in bedbugs? I suggest the authors add a citation to support this idea.

Line 288-289: why are the populations ordered A, C, D and then B when presented for the first time in the methods. I suggest they should be presented in alphabetical order.

Line 287-295: A consider more amount of detail is necessary in this section of the methods. When were the wild populations caught? How long were they maintained in a captive setting before the study was performed? Where were the populations maintained? All in the same location? If so, how might that change the expectations of environmental microbes invading the genital microbiome. Would this standardised housing not homogenise the among-population differences? If so, then I think the paper needs to pay closer consideration of the potential genetic versus environmental impacts of microbes and what factors determine the presence of particular members of a microbiome. How long were they held in a standard environment? Did they all have the same bedding? Was it sterilised before use? Did they all have the same food source? Was it sterilised? I think all of these issues need to be clear in the manuscript. Furthermore, if these conditions were all standard, I think the authors need to think carefully about what they consider an environmental microbiome versus population differences.

Line 304-306: I greatly appreciate the authors methods to prevent contamination, this was very pleasing to see.

Line 313: see general comment below regarding what the genital microbiome is.

Line 317-318: Again, I greatly appreciate the use of sampling controls at this stage in the study to deal with the issue of contamination. This is very pleasing to see in a microbiome study.

Line 321-325: Please provide more details of the DNA extraction approach. Did you follow the kit protocol? It is also unclear if you used a bead beating step. If you did, please make this clear and provide full details of the methods used.

Line 325: Again, great to see the use of controls! Also the use of PCR controls!

Line 329: how many controls run in each library/plate?

Line 341-345: any use of PhiX? Was it 250 bp paired end reads?

Line 359-361: I am very much in support of this approach to deal with contamination. However, I would like to see a bit more information about the contaminants in the supplementary material. What was there? How frequently was it a problem in the controls? Did it affect all controls (e.g. PCR controls, sampling controls, etc.)?

Line 364: It is not clear to me why environmental samples were excluded. What do you mean, any bacteria from environmental sources? Similarly, why exclude uncultured samples? Do you mean these were excluded from the database used for taxonomic assignment? This needs to be made clearer.

Line 381: Which are the internal versus the external reproductive organs? I think this needs to be clearly stated at some point.

Line 398: did you do any model comparison testing to see if the removal of non-significant interactions was a significant improvement to the model.

Line 410: why not use Manhattan distances and PCoA for this analysis instead of creating a presence/absence matrix?

Line 405-417: Given that the analyses presented in this section are presence/absence data I wonder if there could be an issue with 0 inflated data? Did the authors consider this statistical issue in their analysis? Could overdispersion be an issue? Perhaps the authors can comment on this and/or show that this isn't a problem for their data and models.

Line 663: I would recommend trying to make the symbols differentiating the populations more obvious. Currently it is difficult to see the spread of the data very clearly. This comment is also relevant to Figure S1.

Line 696: I think it would be better to have a figure showing the differences between populations (such as the top, left panel of Fig S3) in the main text.

General comments:

1. Defining the genital microbiome. I think the manuscript would benefit from a clearer explanation of what is and what isn't a genital microbiome. For example, I was often unsure if the genital microbiome simply referred to external structures, such as a paramere of males or if this term also included the sperm vesicles.
2. Figure 1 shows a number of different contrasts, and as a result it often becomes difficult to extract the information pointed to in the text (e.g. see comment on line 174-176). I think the authors might consider parsing out the information in Figure 1 into multiple figures.
3. Supplementary tables should have a legend to make sure they can be understood without referring to the main text. Currently they are only briefly mentioned in the main text and this makes it difficult for the reader to follow the flow of information.
4. I wonder if the authors might consider using phylogeny-based methods, such as UniFrac (weighted and/or unweighted), to investigate microbiome composition (presence/absence and abundance). These metrics might show different results that would be well worth knowing, or add further support to the findings of the study.
5. I think it would be beneficial to show the population differences using a PCoA or NMDS plots

Referee: 2

Comments to the Author(s)

In this manuscript, Bellinvia and colleagues report changes in microbial communities in the genital microbiomes of *Cimex lectularius* as a result of mating. The authors report several interesting findings: first, the microbiomes of lab grown *C. lectularius* populations from different geographic locations and collected in different years differ. Second, microbiomes of males and females differ. And third, mating results in changes in genital microbiomes. The authors build on previous reports from the same group; While this is an incremental addition on top of their previous reports, validating the effect of mating on both cuticular and internal microbiomes in different populations collected over different years and locations is of interest. I raise several questions and comments, and suggest a few additional analyses or experiments that would strengthen this manuscript before it is ready for publication.

Introduction:

Generally speaking, the Introduction is very long (~30% of the main text) and often speculative. Speculations and interpretations should be part of the discussion, rather than introduction, and often there is redundancy between the text in the introduction and discussion.

At times the authors make the distinction between sexually transmitted microbes and sexually transmitted opportunistic microbes. While this might be useful in trying to distinguish between pathogens that are sexually transmitted and microbes that happen to be transferred but aren't pathogens, it is confusing and unclear in the text.

Methods: In general, the authors have made an effort to properly control and analyze their data reliably.

Different cultures originated from different geographic locations and were maintained in the lab for different times (9 to over 20 years). How do these populations differ genetically? As the authors note, diet and growth conditions are major drivers in shaping the microbiome and the differences between populations might suggest that a founder effect where initial microbial communities differed significantly, or, alternatively, that genetic differences between these populations affected colonization. Both of these would be of interest, or at least acknowledged. How successful was mating over the 60s given to bugs? Do all mate? Was mating done in a sterile chamber? Were males added before females or vice versa?

Bioinformatic and statistical analyses

How many reads were obtained per sample, or per group on average? Were data subsampled (for beta diversity) or samples with less than a certain number of reads omitted?

The methods report 147 SVs, while Table S3 has 165 SVs.

Nestedness and turnover are interesting metrics. Would also be interesting to use SourceTracker (Knights et al 2011, <https://github.com/biota/sourcetracker2>) to test whether SVs from virgins in each population are source for specific SVs in mated populations.

Why was FDR 1% used for EdgeR?

Results and Discussion:

While 643 samples were sequenced, only 495 samples were analyzed. It is unclear what were the filtering criteria beyond singletons, 0.01% of total reads (how many reads did you collect in total?) and how they affected the data that was left. Generally speaking, losing ~25% of your samples to QC filtration might suggest that the data is not of high quality and that there is an issue with your sequencing method or instrumentation.

This might be a misunderstanding on my part, and I apologize if it is - some of the F statistics reported seems rather low for the reported p values (or q-values). For example $F_{1,177}=2.350$, $q = 0.02$ (which by an F calculator would be 0.13), $F_{3,170}=1.885$, $p = 0.004$ (calculated to be $p=0.12$). In addition, this difference is reported as not significant while <0.05

A direct comparison between populations in virgin bugs would be beneficial in setting up the story. It is mentioned (lines 149-150) but not presented until later. Related to this - the authors should discuss what could be driving microbiome differences in these lab grown populations. Could the experimental setup for mating facilitate bacterial transfer through the media or environment?

It would be useful to report what microbes were found to be transferred by mating, rather than just the numbers of SVs. moreover, it would be useful to compare if the same Genera (or other taxonomic level you are interesting in) are transmitted by mating across populations. At the moment, you are only addressing it by numbers and it is not very informative.

In addition, reporting that few SVs were different between mated and unmated doesn't say much about their relative abundance and how well they were represented.

Author's Response to Decision Letter for (RSPB-2019-2567.R0)

See Appendix A.

RSPB-2020-0302.R0

Review form: Reviewer 1

Recommendation

Accept as is

Scientific importance: Is the manuscript an original and important contribution to its field?

Excellent

General interest: Is the paper of sufficient general interest?

Good

Quality of the paper: Is the overall quality of the paper suitable?

Excellent

Is the length of the paper justified?

Yes

Should the paper be seen by a specialist statistical reviewer?

No

Do you have any concerns about statistical analyses in this paper? If so, please specify them explicitly in your report.

No

It is a condition of publication that authors make their supporting data, code and materials available - either as supplementary material or hosted in an external repository. Please rate, if applicable, the supporting data on the following criteria.

Is it accessible?

Yes

Is it clear?

Yes

Is it adequate?

Yes

Do you have any ethical concerns with this paper?

No

Comments to the Author

The authors have done a really great job revising the manuscript in response to my comments and those of the other reviewer. I think the new analyses significantly strengthen the study and the changes to the wording and formatting make the paper easier to follow. I believe the manuscript will make a strong contribution to our understanding of genital microbiomes and the importance of reproductive microbiomes for evolutionary process, such as sexual selection and reproductive isolation.

Review form: Reviewer 2

Recommendation

Accept as is

Scientific importance: Is the manuscript an original and important contribution to its field?

Good

General interest: Is the paper of sufficient general interest?

Acceptable

Quality of the paper: Is the overall quality of the paper suitable?

Good

Is the length of the paper justified?

Yes

Should the paper be seen by a specialist statistical reviewer?

No

Do you have any concerns about statistical analyses in this paper? If so, please specify them explicitly in your report.

No

It is a condition of publication that authors make their supporting data, code and materials available - either as supplementary material or hosted in an external repository. Please rate, if applicable, the supporting data on the following criteria.

Is it accessible?

Yes

Is it clear?

Yes

Is it adequate?

Yes

Do you have any ethical concerns with this paper?

No

Comments to the Author

I thank the authors for the careful consideration and integration of Reviewer comments to this manuscript. This manuscript will add to the body of knowledge built by this group and others and will surely be of interest to the community.

Decision letter (RSPB-2020-0302.R0)

26-Mar-2020

Dear Ms Bellinva

I am pleased to inform you that your Review manuscript RSPB-2020-0302 entitled "Mating changes the genital microbiome in both sexes of the common bedbug *Cimex lectularius* across

populations" has been accepted for publication in Proceedings B. My apologies for some delay in communicating this decision to you because of the current COVID-19 situation.

The referees have not recommended any further changes. Therefore, please proof-read your manuscript carefully and upload your final files for publication. Because the schedule for publication is very tight, it is a condition of publication that you submit the revised version of your manuscript within 7 days. If you do not think you will be able to meet this date, especially given the current conditions, please let us know immediately.

To upload your manuscript, log into <http://mc.manuscriptcentral.com/prsb> and enter your Author Centre, where you will find your manuscript title listed under "Manuscripts with Decisions." Under "Actions," click on "Create a Revision." Your manuscript number has been appended to denote a revision.

You will be unable to make your revisions on the originally submitted version of the manuscript. Instead, upload a new version through your Author Centre.

1) A text file of the manuscript (doc, txt, rtf or tex), including the references, tables (including captions) and figure captions. Please remove any tracked changes from the text before submission. PDF files are not an accepted format for the "Main Document".

2) A separate electronic file of each figure (tiff, EPS or print-quality PDF preferred). The format should be produced directly from original creation package, or original software format. Please note that PowerPoint files are not accepted.

3) Electronic supplementary material: this should be contained in a separate file from the main text and the file name should contain the author's name and journal name, e.g. `authorname_procb_ESM_figures.pdf`

All supplementary materials accompanying an accepted article will be treated as in their final form. They will be published alongside the paper on the journal website and posted on the online figshare repository. Files on figshare will be made available approximately one week before the accompanying article so that the supplementary material can be attributed a unique DOI. Please see: <https://royalsociety.org/journals/authors/author-guidelines/>

4) Data-Sharing and data citation

It is a condition of publication that data supporting your paper are made available. Data should be made available either in the electronic supplementary material or through an appropriate repository. Details of how to access data should be included in your paper. Please see <https://royalsociety.org/journals/ethics-policies/data-sharing-mining/> for more details.

<http://datadryad.org/submit?journalID=RSPB&manu=RSPB-2020-0302> which will take you to your unique entry in the Dryad repository.

Once again, thank you for submitting your manuscript to Proceedings B and I look forward to receiving your final version. If you have any questions at all, please do not hesitate to get in touch.

Finally, all the best for dealing with the challenges of the COVID-19 situation: I hope you all stay safe.

Yours sincerely,
Professor Loeske Kruuk
mailto:proceedingsb@royalsociety.org

Associate Editor
Board Member
Comments to Author:

The authors have done a good job of responding to the comments raised in the previous round of review, and both reviewers are satisfied, as am I.

Reviewer(s)' Comments to Author:

Referee: 2

Comments to the Author(s).

I thank the authors for the careful consideration and integration of Reviewer comments to this manuscript. This manuscript will add to the body of knowledge built by this group and others and will surely be of interest to the community.

Referee: 1

Comments to the Author(s).

The authors have done a really great job revising the manuscript in response to my comments and those of the other reviewer. I think the new analyses significantly strengthen the study and the changes to the wording and formatting make the paper easier to follow. I believe the manuscript will make a strong contribution to our understanding of genital microbiomes and the importance of reproductive microbiomes for evolutionary process, such as sexual selection and reproductive isolation.

Sincerely,
Proceedings B
mailto: proceedingsb@royalsociety.org

Associate Editor,
Board Member
Comments to Author:

The authors have done a good job of responding to the comments raised in the previous round of review, and both reviewers are satisfied, as am I.

Decision letter (RSPB-2020-0302.R1)

30-Mar-2020

Dear Ms Bellinvia

I am pleased to inform you that your manuscript entitled "Mating changes the genital microbiome in both sexes of the common bedbug *Cimex lectularius* across populations" has been accepted for publication in Proceedings B.

Open Access

Paper charges

Sincerely,

Appendix A

Dear Editor,

We would like to thank you and the two reviewers for the critical evaluation of our manuscript "**Mating changes the genital microbiome in both sexes of the common bedbug *Cimex lectularius* across populations**". We have addressed all comments, which we feel have improved the manuscript a great deal. We have reanalysed the whole data set as suggested and re-produced the figures according to the comments of reviewer 1. For the ease to follow all changes, we have uploaded a highlighted version of the resubmitted manuscript (RSPB-2019-2567) in addition to the normal version.

Kind regards,

Sara Bellinvia

In the following we have replied to each point raised by the reviewers:

Reviewer(s)' Comments to Author:

Referee: 1

Comments to the Author(s)

A recent explosion of studies are demonstrating the importance of the microbiome for host biology. These studies, however, are primarily focused on studies of the gut microbiome, and the majority are on humans or model lab species. In contrast, microbial communities associated with reproductive organs remain relatively unexplored and are rarely considered from an ecological or evolutionary perspective. This is despite the potentially important role these communities may play in reproduction and reproductive processes.

This manuscript presents the results of a study investigating the microbiome of male and female genitalia in the common bedbug, and analyses aimed at testing for differences in microbiomes among populations, between the sexes, and in different reproductive organs, as well as the impact of mating on these genital microbiomes. The authors report distinct microbiomes in the different reproductive organs/cuticle of virgin males and females, though alpha-diversity levels seem not to vary considerably among the sexes, populations, or between mated/virgin individuals. In my opinion, the two most notable results reported in the manuscript are: (1) the finding that mating changed the microbiome composition (community composition and relative abundance of specific SVs) of males and females, and (2) the finding that populations show significant differences in microbiome composition.

In the first instance, some recent studies have demonstrated changes in the microbiome following mating. More broadly, monogamous mating partners appear to share microbes through sexual transmission {Kulkarni:2007ik} and mating can lead to increased similarity in the microbiome of both males and females {Kreisinger:2015bv, White:2010wa, Mandar:2015eu} as well as alterations in these microbial communities {Schoenmakers:2019ds, White:2010wa}. Studies of invertebrates are

generally lacking, however, and as such this study makes a significant contribution to a small, but growing, body of literature. Nonetheless, I do think the paper would benefit from more fully placing the results of this study in the context of the current literature, and thus discussing the results with reference to the above literature both in the introduction and/or discussion text.

We have added literature regarding mating-induced changes in genital microbiomes as suggested by the reviewer (lines 58-59, line 340, line 371).

In addition, I think a discussion whether changes are due to sexual transmission of microbes versus the disruption of the existing microbiome is warranted.

We have clarified that we cannot conclude which process changes the genital microbiome (lines 342-344).

In the second instance, the finding of population differentiation in genital microbiomes is a particularly novel and interesting finding. As the authors of the manuscript discuss, this has very interesting potential ramifications for the evolution of reproductive isolation and speciation. I particularly like this aspect of the study and thus would like to see the results discussed in a little more detail. For example, can then authors perhaps place these findings in the broader context of parasite-mediated speciation.

We have added a discussion section regarding parasite-mediated speciation as suggested by the reviewer (lines 288-294).

Overall, I really liked this study and I think it makes a significant contribution to an emerging field of study on the microbiomes of reproductive systems. However, I think the manuscript would benefit from some additional information and, in some instances, greater clarity in the written text. To facilitate a revision of the manuscript, I have provided a number of comments below (noted by line number) to highlight my concerns or identify text that is unclear.

Line 26: there is a word missing before the text 'specific organ systems' perhaps it should be 'in specific organ systems'

We have changed the sentence as suggested by the reviewer (line 26).

Line 27: I suggest this should be e.g. genital microbiomes, as there are other reproductive organs that harbour microbiomes.

Our intention was to introduce the term to the readers. We have rephrased the sentence to clarify this (line 27).

Line 39: delete 'again'

We have removed the word as suggested by the reviewer.

Line 49: it may be more appropriate to say gastrointestinal microbiome instead of gut in this sentence as the gastrointestinal system is more technically regarded as an organ system.

This sentence was removed due to reformatting of the manuscript.

Line 53: The term intracellular reproductive manipulator is an unfamiliar one. Could the authors use a more familiar term, or perhaps provide an example of an intracellular reproductive manipulator in order to help the reader think about this information.

We have explained the term in more detail (lines 46-47) as suggested by the reviewer.

Line 60/61: I would say the composition of these microbiomes are generally unknown for most species, vertebrates and invertebrates alike. I suggest the authors note this and highlight that this is especially the case for insects.

We have rephrased the sentence as suggested by the reviewer (lines 52-53).

Line 65: I think this section could be reworded somewhat. In the sentence ending on line 65 there is mention of the vaginal microbiome, and yet the next sentence talks about genital microbiomes. If the vaginal microbiome is not a genital microbiome, this needs to be made clear in the definition of the genital microbiome. However, the text suggests they are considered genital microbiomes as the example then given (on line 65/66) concerns the vaginal microbiome.

We have rephrased the paragraph as suggested by the reviewer (lines 54-60).

Line 70: see comment above regarding previous studies demonstrating shifts in microbiome composition with mating.

We have added references as suggested by the reviewer (lines 58-59).

Line 72: can the authors please clarify, what defines an opportunistic microbe (OM)? Is this just a microbe from an environmental source such that the microbiome is a random assemblage of species? Or is it somehow different from a microbe from the environment. The terms seem to be used interchangeably in the manuscript, and as a result it is unclear why this term is used or if it is somehow a subset of microbes that might be from an environmental source. I think a little clarity in what an OM is would be useful for the reader. I think it might also be worth considering if an OM is from an environmental source whether all microbes have an equal opportunity to be an OM or whether there are some processes that influence whether or not an environmental microbe invades or colonises the microbiome of an organ, such as the genitalia.

We have clarified the definition of the term opportunistic microbe as suggested by the reviewer (lines 74-76). We have added factors that might affect the invasion of OM (lines 64-68).

Line 75: why might OM have a more direct effect compared to STM?

We have rephrased the sentences to clarify why OM might have a more direct effect (lines 70-72).

Line 82: It is not clear to me why the example of intestinal microbiome of mice is used as an example of sexually transmitted microbes disturbing the resident microbiome.

We agree with the reviewer and have replaced the example of the intestinal microbiome with a study on the vaginal microbiome in humans (lines 74-75).

Line 94: I think ethnicity is a more correct term than ancestry. In the original publication it is mentioned as ethnicity. I think this supports the idea of population level variation in microbiomes more clearly.

We agree with the reviewer and have rephrased the sentence as suggested (line 84).

Line 116/117: please provide a citation for the sentence that microbial communities differ between organs and mating increases microbiome similarity between males and females. More generally, I think the authors previous work could be explained in more detail in order to clarify the aims of the current study and to help distinguish between the findings of the previous publication and this current paper. I think this will more clearly set out what is known in the system and what this paper adds to our knowledge.

We have rephrased the paragraph as suggested by the reviewer and added information about the additional value of this study (lines 96-107).

Line 123/124: is it differences in external environment or population level variation that is being investigated. I thought the latter.

We have clarified this by rephrasing the sentence (lines 105-106).

Line 125: by groups, do you mean populations? Please clarify.

This sentence was removed due to reformatting of the manuscript.

Line 136: Given that results are presented first, I think it needs to be made clear what reproductive organs are being investigated. More generally, in the text it is frequently unclear what an internal versus external organ is. I think the authors could present the information on what organs were investigated and whether these are internal or external in a more clear and explicit way.

We have moved the methods section before the results and discussion section. Furthermore, we have added two sentences that clarify which organs were investigated and which organs are internal or external (lines 208-210, lines 242-244).

Line 143: differences between cuticle and genital microbiomes are not always significant, thus I think the first sentence of this results section is not true. Perhaps this text could be reworded.

We have rephrased the paragraph as suggested (lines 251-257).

Line 143-148: given that the results are presented first, I think it is important to tell the reader how microbiomes were assessed. For example, 16S rRNA amplicon sequencing was used to investigate the structure of the microbial communities.

We have added information on the sequencing method as suggested by the reviewer (lines 240-241).

Line 145-147: q instead of p in p-values.

We have specified why we use q-values instead of p-values (correction for multiple comparisons, line 254).

Line 144-147: the text states ‘...their relative abundance.....but not between the cuticle and the external..... or the internal reproductive organs of both sexes and the paramere.....’ but the p values stated is 0.02, which suggests a significant difference. Thus, either the text or the statistics reported seem to be in error.

We agree with the reviewer and have rephrased the sentence (lines 255-256).

Line 147: what ‘groups’?

We have rephrased the sentence to clarify which groups were compared (lines 257).

Line 148: how was between-individual difference estimated? I know this is in the methods, but not including it here means the reader has to flip back to the methods to understand the results. Thus, I think the authors should makes these measures clear in the results section.

We have added information about the statistical tests in the results section as suggested by the reviewer.

Line 150:by ‘location’ do you mean site on the host? Or population location?

We have rephrased the sentence to clarify that we meant organ location instead of population location (lines 260-261).

Line 156: by ‘other organs’ do the authors mean all other organs investigated. At this point I had kind of lost track of the different organs being investigated, so I think it would help the reader to be explicit here.

Line 157: can the authors clarify what they mean by the function of the organ driving the finding?

We have removed both phrases since the dataset changed after we had corrected a mistake in the quality filtering step that excludes sequences with less than 0.01% of the reads. The corrected dataset showed no significant difference between paramere and cuticle.

Line 161: which genital microbiomes? All? Just males? Females? If they are all lumped, is this appropriate?

We agree with the reviewer and we have rephrased the sentence to distinguish between female and male microbiomes (lines 269-270).

Line 162: again, what is meant by 'group' here.

We have removed this part of the sentence as it gives the same information as the following sentences.

Line 164: Is the comparison of the sexes mentioned here for all organs investigated lumped together?

Yes, this comparison includes all organs within the given sex. Organs within sex are compared later on (lines 273-275).

**Line 164: I advise the use of panels (A-D) on Figure S3 to clarify to the reader which graph to look at.
Line 165: Should the results of different organs from the same sex also point to figure S3?**

We have added labels to the figure and referenced them in the text (lines 272-275).

Line 169: Are these then the most prevalent of the SVs identified in the study?

Yes, they are. We have rephrased the sentence to clarify this (line 280).

Line 172: any ideas for why this Rickettsia strain was more prevalent in females?

We think there are several possible reasons. It might fulfil a specific function in the genital microbiome, i.e. be a mutualist, be transmitted horizontally via the ovaries, and / or infect a specific organ in females.

Line 174-176: I think it would be useful to also show the actual number of shared SVs relative to the total number of SVs (e.g. 44/50) in addition to the % as the information in Figure 1 extends beyond that of the differences between virgin males and females.

We have added the numbers as suggested by the reviewer (lines 286-288).

Line 182-189: I greatly appreciate that the authors do not want to speculate to widely about their findings. However, I think it would be useful here to discuss species and population-specific

microbiomes more broadly (i.e. discuss microbiomes of a range of host sites). For example, a number of studies have shown variation in microbiomes in relation to host phylogeny.

We have added a metagenomics study on bedbug microbiomes across sampling sites (lines 298-299) as suggested by the reviewer. We have also added an explanation how these differences could arise (lines 300-308).

Line 204: the sub heading here mentioned relative abundance of SV, but it is a little confusing with the subheading on line 248 which deals with abundance changes. Perhaps the authors could change the wording to make the headings clearer and the difference between them more explicit.

We have rephrased the subheading as suggested (line 322).

Line 206: As before, I think it is necessary to indicate how these data were generated and how specific metrics estimated.

We have added information about statistical tests. We think that it is unnecessary to explain the sequencing method again because it is mentioned in the introductory paragraph of the results section.

Line 207: which part of figure S3 is referenced here? Panels would be useful.

We have added panels to the figure and referenced them in the text (Figure 1, line 325).

Line 213: Which two out of four populations?

We have added the information about the two populations (line 331).

Line 216: replace 'q' with 'p'

We have adjusted the p-values with the Benjamini-Hochberg procedure. Therefore, it should be q instead of p.

Line 228: Not being familiar with this method, I found the results/discussion and the methods section on the analysis of nestedness and turnover very difficult to follow. I suggest that the authors explain this methodology in greater detail, including some mention of why these processes are expected to account for changes in the microbiome. It is difficult to understand exactly how strain replacement differs from strain loss/introduction.

We have replaced the nestedness and turnover metrics with results from SourceTracker2 because we think this method is better suited for analysing sexual transmission.

Line 249: I think the authors might consider more graphical ways in which to display their data. For example, I think the change in microbiome composition (what is present and the abundance of the different SVs) could be more clearly conveyed using a heatmap approach.

We have added a heatmap showing prevalence and abundance changes as suggested by the reviewer (Figure 4).

Line 257-258: I think this is a very interesting result (ie. that abundance change did not vary by sex). However, I think the idea that females are expected to be more affected by mating needs to be more clearly laid out and supported by current literature.

Line 260: Can bacteria more readily enter the female body? Your results suggest this is perhaps not the case in bedbugs? I suggest the authors add a citation to support this idea.

We have added an explanation for our prediction and a citation supporting this prediction (lines 370-372).

Line 288-289: why are the populations ordered A, C, D and then B when presented for the first time in the methods. I suggest they should be presented in alphabetical order.

We agree with the reviewer and have changed the labelling as suggested (lines 112-113).

Line 287-295: A consider more amount of detail is necessary in this section of the methods. When were the wild populations caught? How long were they maintained in a captive setting before the study was performed? Where were the populations maintained? All in the same location? If so, how might that change the expectations of environmental microbes invading the genital microbiome. Would this standardised housing not homogenise the among-population differences? If so, then I think the paper needs to pay closer consideration of the potential genetic versus environmental impacts of microbes and what factors determine the presence of particular members of a microbiome. How long were they held in a standard environment? Did they all have the same bedding? Was it sterilised before use? Did they all have the same food source? Was it sterilised? I think all of these issues need to be clear in the manuscript. Furthermore, if these conditions were all standard, I think the authors need to think carefully about what they consider an environmental microbiome versus population differences.

We have added more details on the housing conditions (lines 114-117). Even though all bedbug populations are kept in washed containers and get the same filter paper, bacterial strains on filter papers originating from the faeces are population-specific as mentioned by Otti et al. (2017, *Frontiers in Immunology*). These bacteria might be taken up by the paramere and the cuticle of males and females, finally being transferred to the genital microbiomes. We are aware that population-specific genital microbiomes might also be caused by bacterial strains adapted to host genotype. Therefore, we recommend to conduct an experiment with isogenic lines of different genotypes, expose them to different communities of environmental microbes and sequence the genital microbiome. If genital microbiomes differ between isogenic lines exposed to different microbes, this would suggest that environmental microbes shape genital microbiomes. If genital microbiomes exposed to the same environmental community differ between genotypes, this would hint at an adaptation of microbiomes to host genotype.

Line 304-306: I greatly appreciate the authors methods to prevent contamination, this was very pleasing to see.

Thank you.

Line 313: see general comment below regarding what the genital microbiome is.

Line 317-318: Again, I greatly appreciate the use of sampling controls at this stage in the study to deal with the issue of contamination. This is very pleasing to see in a microbiome study.

Thank you.

Line 321-325: Please provide more details of the DNA extraction approach. Did you follow the kit protocol? It is also unclear if you used a bead beating step. If you did, please make this clear and provide full details of the methods used.

We have rephrased the paragraph as suggested by the reviewer (lines 146-148).

Line 325: Again, great to see the use of controls! Also the use of PCR controls!

Thank you.

Line 329: how many controls run in each library/plate?

We have added information on the number of controls (lines 167-170).

Line 341-345: any use of PhiX? Was it 250 bp paired end reads?

We have added the missing information (lines 166-167).

Line 359-361: I am very much in support of this approach to deal with contamination. However, I would like to see a bit more information about the contaminants in the supplementary material. What was there? How frequently was it a problem in the controls? Did it affect all controls (e.g. PCR controls, sampling controls, etc.)?

We have added a table showing all contaminants and their prevalence in each control type (Table S2).

Line 364: It is not clear to me why environmental samples were excluded. What do you mean, any bacteria from environmental sources? Similarly, why exclude uncultured samples? Do you mean these were excluded from the database used for taxonomic assignment? This needs to be made clearer.

We have clarified the method by rephrasing the sentence (line 188-189).

Line 381: Which are the internal versus the external reproductive organs? I think this needs to be clearly stated at some point.

We have added information on the given organs (lines 208-210) as suggested by the reviewer.

Line 398: did you do any model comparison testing to see if the removal of non-significant interactions was a significant improvement to the model.

We calculated the AIC and found that removing non-significant interactions improved the model fit.

Line 410: why not use Manhattan distances and PCoA for this analysis instead of creating a presence/absence matrix?

We have removed this analysis because we think SourceTracker is more suitable for analysing sexual transmission.

Line 405-417: Given that the analyses presented in this section are presence/absence data I wonder if there could be an issue with 0 inflated data? Did the authors consider this statistical issue in their analysis? Could overdispersion be an issue? Perhaps the authors can comment on this and/or show that this isn't a problem for their data and models.

We have clarified that the abundance tests were conducted on read numbers and not on presence-absence data (line 201, line 230). Furthermore, before abundances were subjected to edgeR, a pseudo-count of 1 was added to avoid zero inflation (line 232).

Line 663: I would recommend trying to make the symbols differentiating the populations more obvious. Currently it is difficult to see the spread of the data very clearly. This comment is also relevant to Figure S1.

We have changed the figure as suggested by the reviewer (Figure S1). We have replaced the figure showing the log₂fold change with a heatmap that shows mating-induced changes in prevalence and abundance (Figure 4).

Line 696: I think it would be better to have a figure showing the differences between populations (such as the top, left panel of Fig S3) in the main text.

We have moved the PCoA plot to the main text as suggested by the reviewer (Figure 1).

General comments:

- 1. Defining the genital microbiome. I think the manuscript would benefit from a clearer explanation of what is and what isn't a genital microbiome. For example, I was often unsure if the genital microbiome simply referred to external structures, such as a paramere of males or if this term also included the sperm vesicles.**

We have added a definition for genital microbiomes in bedbugs as suggested by the reviewer (line 102.103).

- 2. Figure 1 shows a number of different contrasts, and as a result it often becomes difficult to extract the information pointed to in the text (e.g. see comment on line 174-176). I think the authors might consider parsing out the information in Figure 1 into multiple figures.**

We have highlighted the important numbers (Figure 2). We think that splitting the figure into multiple figures will not give the whole picture required to understand the distribution of microbes across sexes and populations.

- 3. Supplementary tables should have a legend to make sure they can be understood without referring to the main text. Currently they are only briefly mentioned in the main text and this makes it difficult for the reader to follow the flow of information.**

We have separated the supplementary section from the main text and we have added an index giving all legends.

- 4. I wonder if the authors might consider using phylogeny-based methods, such as UniFrac (weighted and/or unweighted), to investigate microbiome composition (presence/absence and abundance). These metrics might show different results that would be well worth knowing, or add further support to the findings of the study.**

We have added statistical analyses of weighted UniFrac distances for comparisons between cuticular and genital microbiomes (results: line 252 and line 259) and between populations, sexes, and organs (results: lines 271-279) as suggested by the reviewer. We did not use UniFrac for the statistical comparison between virgin and mated microbiomes because we do not think it would be meaningful. If communities differ between virgin and mated individuals, this difference due to introduced or lost bacterial strains as well as due to changes in host physiology should not depend on the relatedness of present strains. Evolutionary change would not be possible during the small time period between mating and dissection and hence it is unlikely that closely related bacterial strains in mated individuals originate from an ancestral strain in virgin individuals.

- 5. I think it would be beneficial to show the population differences using a PCoA or NMDS plots**

We have moved the PCoA plot to the main text (Figure 1).

Referee: 2

Comments to the Author(s)

In this manuscript, Bellinva and colleagues report changes in microbial communities in the genital microbiomes of *Cimex lectularius* as a result of mating. The authors report several interesting findings: first, the microbiomes of lab grown *C. lectularius* populations from different geographic locations and collected in different years differ. Second, microbiomes of males and females differ.

And third, mating results in changes in genital microbiomes. The authors build on previous reports from the same group; While this is an incremental addition on top of their previous reports, validating the effect of mating on both cuticular and internal microbiomes in different populations collected over different years and locations is of interest. I raise several questions and comments, and suggest a few additional analyses or experiments that would strengthen this manuscript before it is ready for publication.

Introduction:

Generally speaking, the Introduction is very long (~30% of the main text) and often speculative. Speculations and interpretations should be part of the discussion, rather than introduction, and often there is redundancy between the text in the introduction and discussion.

We have shortened the introduction and removed speculative sentences in the introduction as suggested by the reviewer.

At times the authors make the distinction between sexually transmitted microbes and sexually transmitted opportunistic microbes. While this might be useful in trying to distinguish between pathogens that are sexually transmitted and microbes that happen to be transferred but aren't pathogens, it is confusing and unclear in the text.

Since we are not only interested in sexual transmission of bacteria but also whether these are typical bacteria causing STI or opportunistic microbes, for instance from the cuticle, we have to distinguish between these. We have tried to make the distinction clearer by adding a better definition for opportunistic microbes (lines 64-66).

Methods:

In general, the authors have made an effort to properly control and analyze their data reliably.

Thank you.

Different cultures originated from different geographic locations and were maintained in the lab for different times (9 to over 20 years). How do these populations differ genetically? As the authors note, diet and growth conditions are major drivers in shaping the microbiome and the differences between populations might suggest that a founder effect where initial microbial communities differed significantly, or, alternatively, that genetic differences between these populations affected colonization. Both of these would be of interest, or at least acknowledged.

We do not know how the populations used in our study differ genetically. However, previous research has shown that bedbugs from infestations within the same city (London) exhibit strong genetic differentiation (Meriweather et al. 2013, PLoS One). We have added a section dealing with this topic (lines 296-308).

How successful was mating over the 60s given to bugs? Do all mate? Was mating done in a sterile chamber? Were males added before females or vice versa?

Normally, virgin bedbugs immediately mate. In case a mating was not successful, both male and female were discarded. We have clarified this in line 129. The mating was not done in a sterile chamber since none of the bedbug populations were held in a sterile chamber and sexual transmission should not be affected by microbes in the air. Females were added before males.

Bioinformatic and statistical analyses

How many reads were obtained per sample, or per group on average? Were data subsampled (for beta diversity) or samples with less than a certain number of reads omitted?

We have included information about the average number of reads (lines 246-247) as suggested by the reviewer. We did not subsample data after the mentioned filtering steps.

The methods report 147 SVs, while Table S3 has 165 SVs.

The number of SVs is the same for the text and Table S3. All SVs were given a number before the filtering step so that some numbers got lost during filtering. Since we have corrected a mistake in the quality filtering step that excludes sequences with less than 0.01% of the reads, the number of SVs has decreased from 147 to 126 SVs.

Nestedness and turnover are interesting metrics. Would also be interesting to use SourceTracker (Knights et al 2011, <https://github.com/biota/sourcetracker2>) to test whether SVs from virgins in each population are source for specific SVs in mated populations.

Thank you for your advice, which we feel has improved our data analysis a lot. We have replaced the nestedness and turnover analysis because in our opinion, SourceTracker is more suitable for answering our question.

Why was FDR 1% used for EdgeR?

This arbitrary threshold was chosen to obtain a more conservative result. To make it easier for the reader, we have change the FDR to 5% (line 233).

Results and Discussion:

While 643 samples were sequenced, only 495 samples were analyzed. It is unclear what were the filtering criteria beyond singletons, 0.01% of total reads (how many reads did you collect in total?) and how they affected the data that was left. Generally speaking, losing ~25% of your samples to QC filtration might suggest that the data is not of high quality and that there is an issue with your sequencing method or instrumentation.

We have added the initial read numbers and the ones after all filtering steps (lines 184-186). We only removed singletons, contaminants and taxa with less than 0.01% of total reads. We are aware that we lost a high proportion due to QC filtration. However, since the amount of tissue used for sequencing was very low, it is not surprising that we have a higher probability of sequencing mistakes and

contamination compared to microbiomes from other tissues. We took the risk of losing several reads because we wanted to be able to analyse mating effects, which would have been difficult when pooling individuals.

This might be a misunderstanding on my part, and I apologize if it is - some of the F statistics reported seems rather low for the reported p values (or q-values). For example $F_{1,177}=2.350$, $q = 0.02$ (which by an F calculator would be 0.13), $F_{3,170}=1.885$, $p = 0.004$ (calculated to be $p=0.12$).

In comparison to F statistics from an ANOVA, a PERMANOVA computes pseudo-F statistics for each permutation and compares them to the unpermuted statistic. Therefore, p-values from a PERMANOVA are not directly comparable to p-values from an ANOVA. We have clarified that the F values in the text were not obtained with an ANOVA but a PERMANOVA to avoid confusion about the statistics (e.g. line 292).

In addition, this difference is reported as not significant while <0.05

We have corrected the mistake (lines 255-256).

A direct comparison between populations in virgin bugs would be beneficial in setting up the story. It is mentioned (lines 149-150) but not presented until later. Related to this - the authors should discuss what could be driving microbiome differences in these lab grown populations.

We have clarified that the word "location" did not refer to sample origin but to body site (line 260-261). This part of the analysis was not based on population differences. We have added a possible explanation for population differences (lines 296-308) as suggested by the reviewer.

Could the experimental setup for mating facilitate bacterial transfer through the media or environment?

Since we used fresh filter paper for each mating, we do not think that the experimental setup could have facilitated bacterial transfer. It is more likely that the setup weakened the transfer. In a natural environment, bedbugs would mate in their refuges that could be contaminated with many – probably mostly fecal- bacteria species that should not be present on the filter paper that was changed after each mating.

It would be useful to report what microbes were found to be transferred by mating, rather than just the numbers of SVs. moreover, it would be useful to compare if the same Genera (or other taxonomic level you are interesting in) are transmitted by mating across populations. At the moment, you are only addressing it by numbers and it is not very informative. In addition, reporting that few SVs were different between mated and unmated doesn't say much about their relative abundance and how well they were represented.

We have added information on the transmitted microbes in table 1 and table 2 and in the results section (lines 332-337) as suggested by the reviewer.